# Multi-omics analysis reveals COVID-19 vaccine induced attenuation of inflammatory responses during breakthrough disease

The immune mechanisms mediating COVID-19 vaccine attenuation of COVID-19 remain undescribed. We conducted comprehensive analyses detailing immune responses to SARS-CoV-2 virus in blood post-vaccination with ChAdOx1 nCoV-19 or a placebo. Samples from randomised placebo-controlled trials (NCT04324606 and NCT04400838) were taken at baseline, onset of COVID-19-like symptoms, and 7 days later, confirming COVID-19 using nucleic amplification test (NAAT test) via real-time PCR (RT-PCR). Serum cytokines were measured with multiplexed immunoassays. The transcriptome was analysed with long, short and small RNA sequencing. We found attenuation of RNA inflammatory signatures in ChAdOx1 nCoV-19 compared with placebo vaccinees and reduced levels of serum proteins associated with COVID-19 severity. *KREMEN1*, a putative alternative SARS-CoV-2 receptor, was downregulated in placebo compared with ChAdOx1 nCoV-19 vaccinees. Vaccination ameliorates reductions in cell counts across leukocyte populations and platelets noted at COVID-19 onset, without inducing potentially deleterious Th2-skewed immune responses. Multi-omics integration links a global reduction in miRNA expression at COVID-19 onset to increased pro-inflammatory responses at the mRNA level. This study reveals insights into the role of COVID-19 vaccines in mitigating disease severity by abrogating pro-inflammatory responses associated with severe COVID-19, affirming vaccine-mediated benefit in breakthrough infection, and highlighting the importance of clinically relevant endpoints in vaccine evaluation.

Vaccines have been instrumental in preventing morbidity and mortality from COVID-19. Within the first year of their implementation, an estimated 14 to 18 million deaths were averted through vaccination[1]. As of February 2023, over 5.5 billion people have received at least one COVID-19 vaccine, altering the course of the pandemic[2]. The success of COVID-19 vaccine programmes belies initial uncertainty surrounding when, or even if, an effective vaccine would be found. Failure rates in vaccine development are very high; some pathogens still have no licensed vaccines despite decades of work (e.g., HIV and *Staphylococcus aureus*). Furthermore, preclinical animal studies of vaccines against other coronavirus have raised theoretical safety concerns related to certain types of vaccines that could result in non-protective and potentially detrimental or disease-enhancing immune responses following coronavirus exposure[3,4]. However, this adverse phenomenology has not subsequently been observed in preclinical, clinical or post-implementation vaccine safety data of COVID-19 vaccines[5–11].

e-mail: daniel.oconnor@paediatrics.ox.ac.uk

Although SARS-CoV-2 continues to circulate, with new variants emerging, breakthrough COVID-19 in vaccinated individuals is typically milder than in immunologically naïve unvaccinated individuals, despite viral loads sometimes being similar between the groups[12,13]. The immune mechanisms mediating disease attenuation are poorly understood, and this lack of understanding makes optimal vaccination policies (e.g., booster vaccine campaigns) aimed at minimising illness severity in the post-vaccination era challenging.

Recent studies use a plethora of contemporary methodologies to provide a detailed picture of the immunological responses to SARS-CoV-2 at varying disease severities[14–17]. However, there is a paucity of detailed evaluation of how vaccination modifies these immunological responses in the absence of sterilising immunity[18]. The phase 1 to 3 randomised single-blind controlled trials of the ChAdOx1-nCoV-19 vaccine (AZD1222) offer a unique opportunity to explore how ChAdOx1 nCoV-19 vaccination influences the immune response during breakthrough infection[6,19,20]. Here we analyse the whole-blood coding, non-coding and small-noncoding transcriptome, and serum cytokines and chemokines to give a high-resolution description of immune responses during COVID-19 in ChAdOx1 nCoV19 or placebo (MenACWY) vaccinated participants. We also compare COVID-19 immune responses to non-covid infections with similar symptomology to give further insights into COVID-19-specific immune responses. Our study characterises the beneficial immune response conferred by ChAdOx1 nCoV-19 vaccine during breakthrough infection at the fine molecular level. The study addresses a knowledge gap surrounding vaccine-mediated attenuation of infectious disease.

This information is a step towards optimal vaccination policies aimed at minimising illness severity associated with COVID-19 and a rationalised approach for future vaccine development and implementation (e.g., robust correlates of protection against severe disease). Moreover, molecular markers associated with disease severity may form targets for future drug development or therapies to augment vaccine-mediated protection.

## Results

### Multi-omics analysis of COVID-19 in a randomised controlled vaccine trial – cohort description

During the phase 2/3 trials of the ChAdOx1 nCoV-19, we collected samples to capture the molecular response to COVID-19 in participants experiencing specific symptoms associated with COVID-19. All symptomatic COVID-19 cases included here occurred between June and December 2020 in vaccinated participants of the phase 2/3 trials of ChAdOx1 nCoV-19. In stage 1, we analysed samples from 16 symptomatic COVID-19 episodes (i.e. NAAT +ve episodes, 7 ChAdOx1 nCoV-19 vaccinees and 9 placebo vaccinees) and 17 NAAT-ve (8 ChAdOx1 nCoV-19 and 9 placebo vaccinees) episodes—those with COVID-19-like symptoms who did not have COVID-19 (Fig. 1a). This stage enabled comparison of COVID-19 compared with non-COVID-19 episodes with similar symptomology. Baseline samples from 11 participants were available for comparison within this cohort.

Stage 2 was designed to confirm distinct molecular responses to COVID-19 between ChAdOx1 nCoV-19 and placebo vaccinees in a larger independent cohort; thus, it was restricted to NAAT+ve episodes, of whom 21 received the ChAdOx1 nCoV-19 vaccine, and 30 received the placebo vaccine (MenACWY) (Fig. 1, Supplementary Table 5 and Supplementary Table 6). Nineteen samples from subjects matched for risk factors were included in stage 2 (Fig. 1b). Supplementary Table 3 and Supplementary Table 4 details the assays run per group and timepoint in each stage. Demographics and risk factors for severe COVID-19 were similar across all groups (Supplementary Table 5 and Supplementary Table 6). All COVID-19 episodes were mild or moderate (Supplementary Table 1, Supplementary Table 5 and Supplementary Table 6). Full demographic information about the participants, samples and time points collected for each of the omics datasets is available in Supplementary data 1. RNA-seq statistics e.g. average number of reads per sample, read length are contained in Supplementary Table 2.

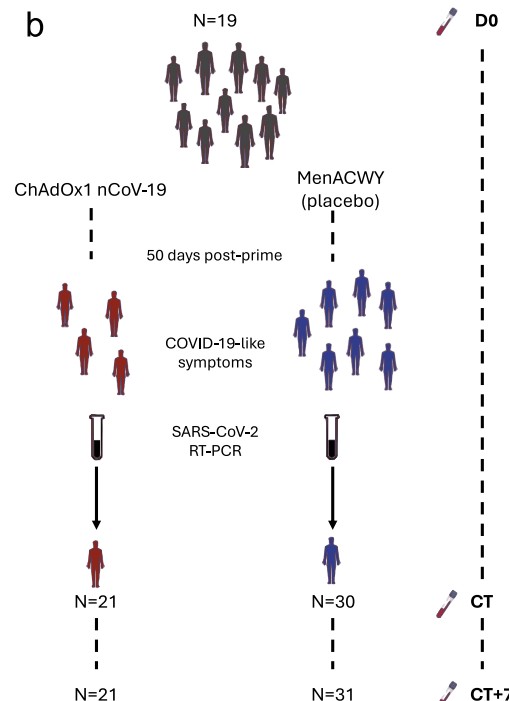

**Fig. 1 | Study overview and cohort description. a** Stage 1, symptomatic cohort: ChAdOx1 nCoV-19 vaccinees are shown in red. MenACWY vaccinees are placebo vaccinees (blue). NAAT+ve are participants with COVID-19, NAAT-ve are people who had COVID-19-like symptoms and thus had a CT visit but were subsequently shown not to have COVID-19 (grey); (**b**) Stage 2: all participants had COVID-19 (i.e. were NAAT+ve). D0: Baseline visit, before vaccination. CT: COVID-19 test visit. CT + 7: 7 days after CT visit.

## Distinct molecular signatures in acute COVID-19 compared with healthy and symptomatic controls in stage 1 results

In the stage 1, we investigated the distribution and relationships between samples from different study groups (Fig. 2a). Gene set enrichment analysis revealed significantly enriched pathways between COVID-19 cases, symptomatic controls (i.e. COVID-19 negative cases), and health (baseline) (Supplementary Fig. 1a–f), reflecting distinct next-gen RNA blood transcriptome profiles. The time since the last vaccination until CT was not affecting the clustering of NAAT+ve ChAdOx1 nCoV-19 (Supplementary Fig. 1i). There were 1,119 differentially expressed genes (DEGs) at CT between NAAT-ve participants and NAAT+ve placebo vaccinees (Fig. 2b), with GSEA revealing positive enrichment of terms such as "response to virus" in NAAT+ve placebo vaccinees (Fig. 2c). Conversely, no DEGs or enriched pathways were observed at CT between NAAT-ve participants and NAAT+ve ChAdOx1 nCoV-19 vaccinees (Supplementary Fig. 1g). We employed long-read 3rd gen RNA sequencing to improve identification of gene isoforms and enable differential transcript expression analysis. In-line with the gene expression data, differences in transcripts were observed between CT samples from NAAT+ve episodes and CT samples from NAAT-ve individuals and baseline samples (Fig. 2 and Supplementary Fig. 3a). There were no differences in transcriptome between ChAdOx1 nCoV-19 vaccinees and Placebo NAAT-ve groups at CT (Supplementary Fig. 4).

Small RNA (sRNA) features were differentially expressed between NAAT+ve placebo vaccinees and NAAT-ve participants at CT (Fig. 2d). These sRNA features included microRNAs (miRNAs) and sRNAs not derived from miRNAs (e.g., tRNA fragments). Consistent with the gene expression results, no sRNAs were differentially expressed at CT between NAAT-ve participants and ChAdOx1 nCoV-19 vaccinees (Supplementary Fig. 1h). Furthermore, the magnitude of $\log_2$ fold-changes and effect sizes were also smaller in the ChAdOx1 nCoV-19 group vs the placebo group when compared with the NAAT-ve group in the sRNA data (Supplementary Fig. 20). MiRNA enrichment analysis suggested increased targeting of immune pathways in NAAT-ve participants compared with participants with COVID-19, potentially identifying a paucity of post-transcriptional regulation in COVID-19 infection compared with other infections or illnesses (Supplementary Fig. 10 and Supplementary Data 7). Serum levels of IP10 and IL10 were higher at CT in COVID-19 cases compared with NAAT-ve participants, regardless of vaccine group (Fig. 2e and Supplementary fig. 2) (IP10: fdr = 0.026 compared to NAAT+ve ChAdOx1 nCoV-19 vaccinees and fdr = 0.0001 compared to NAAT+ve placebo vaccinees; fdr = 0.003 and 0.0006 for IL10, respectively). An increase in MCP1 was also seen, but this observation was restricted to NAAT+ve placebo vaccinees (Fig. 2e, fdr = 0.001).

## Broad resolution of COVID-19 gene signature seven days post-symptom onset but enrichment of humoral immune responses

Seven days after participants presented with COVID-19 symptoms (CT + 7), global gene expression trajectories returned towards baseline with CT + 7 samples predominantly moving to the bottom of the plot compared to CT samples and located in the proximity to D0 (Fig. 3a).

There were no DEGs between CT + 7 and baseline in either of the study groups, demonstrating a resolution of transcriptome perturbation over time (Supplementary Fig. 24a–c). Despite resolution in differential gene expression at CT + 7, GSEA revealed positive enrichment of genes associated with humoral immune responses and regulation of complement (Fig. 3b, c). Resolution of perturbation was less evident in the sRNA sequencing data with DEGs observed in NAAT+ve placebo vaccinees at day 7 (Fig. 3d), however, no DEGs in ChAdOx1 nCoV-19 (Supplementary Fig. 24d) compared with baseline. Serum cytokine levels in both vaccine arms had returned to baseline levels by CT + 7 (Supplementary Fig. 13).

## ChAdOx1 nCoV-19 attenuates the molecular response to SARS-CoV-2 breakthrough infection

In this study's first stage, we observed distinct molecular responses to SARS-CoV-2 infection between ChAdOx1 nCoV-19 vaccinated individuals and placebo vaccinees (Fig. 4). At COVID-19 onset, 684 DEGs were observed in the placebo vaccine arm, compared with baseline (Fig. 4a). Despite general agreement in gene regulation at COVID-19 onset amongst the two vaccine groups compared with baseline, gene perturbation was attenuated in the ChAdOx1 nCoV-19 group (Fig. 4a, b). Direct comparison of the ChAdOx1 nCoV-19 and placebo vaccine groups at CT identified 5 DEGs (Fig. 4c). GSEA of this comparison showed enrichment of pathways such as response to virus and type I interferons in the placebo vaccine group, while pathways related to formation of proteins in the cytoplasm were upregulated in the ChAdOx1 nCoV-19 group (Supplementary Fig. 1f). Similar findings were also seen at the transcript level (3rd-gen ONT RNA-seq) and sRNA data (Fig. 4d, e, h and Supplementary Figs. 18, 20). Five sRNAs were differentially expressed in ChAdOx1 nCoV-19 vaccinees compared with placebo vaccinees at CT (Fig. 4f). In addition, serum levels of IFN-γ and IP10 were higher at CT in the placebo vaccinees compared with ChAdOx1 nCoV-19 vaccinees (Fig. 4g) (fdr = 0.045 and 0.034, respectively). 3rd-gen ONT RNA-seq revealed four differentially expressed transcripts (DETs) (Fig. 4h, Supplementary fig. 3b), including an isoform of MX1 gene, MX1-201, which was upregulated at CT in placebo vaccines (Fig. 4i, D0 vs NAAT+ve placebo fdr = 0.00345; NAAT-ve vs NAAT+ve placebo fdr = 0.003108; NAAT+ve ChAdOx1 vs NAAT+ve placebo fdr = 0.018). This isoform results in a shorter (654 aa) protein product with the deletion of 8 aa (Met479-Asp486) that forms part of the α-helix of the stalk region of the protein that is responsible for the oligomerisation and function[21] (Supplementary Fig. 3c, d).

## ChAdOx1 nCoV-19 attenuation of the molecular response to SARS-CoV-2 infection is confirmed in an independent cohort

We further explored vaccine-related molecular differences in the immune response during COVID-19 in a larger, independent cohort – stage 2 of the study (Fig. 5). As before, there was general agreement in gene regulation at COVID-19 onset amongst the two vaccine groups compared with baseline; however, once again, gene perturbation was attenuated in the ChAdOx1 nCoV-19 group (Fig. 5a, e), with more DEGs between CT and baseline in placebo vaccinees than in ChAdOx1 nCoV-19 vaccinees; 5,644 vs 1,538 DEGs respectively (Fig. 5b, c). The time between last vaccine dose and CT does not affect clustering of NAAT+ve ChAdOx1 nCoV-19 (Supplementary Fig. 5b) – this is similar to the stage 1 cohort. As in stage 1 cohort, there were differences between vaccine groups at COVID-19 onset, with 1,131 genes differentially expressed between ChAdOx1 nCoV-19 and placebo vaccine recipients at CT. GSEA revealed enrichment of pathways such as type I interferon and complement activation (Fig. 5f and Supplementary Fig. 5a). ChAdOx1 nCoV-19 associated attenuation in perturbation was also seen in the sRNA analysis with more differentially expressed sRNAs at COVID-19 onset versus baseline samples in placebo vaccinees compared with ChAdOx1 nCoV-19 vaccinees; 410 sRNAs versus 307 sRNAs respectively (Fig. 6a, b) with absolute foldchanges and effect sizes of those differentially expressed sRNAs being larger in the placebo group compared with the ChAdOx1 nCoV-19 group (Supplementary Fig. 21). Differences in serum cytokine concentrations were again observed at COVID-19 onset between the placebo and vaccinees, with higher levels of IFN-γ, IL18, IP10, IL10 and TNF-α in placebo vaccinees (Fig. 6c–e) (fdr = 0.042 and 0. 019 for IFN- γ, 0.034 for IL18, 0.0035 and 0.0044 for IP10, 0.013 and 0.013 for IL10, 0.0057 and 0.0057 for TNF-α at CT and CT + 7 respectively). We did not observe any differences in Th2 serum cytokines between vaccine groups during COVID-19 (Supplementary Fig. 6). To assess if this "molecular attenuation" was mediated by viral load, the minimum cycle threshold values from the SARS-CoV-2 RT-PCR during the COVID-19 episode were compared between the

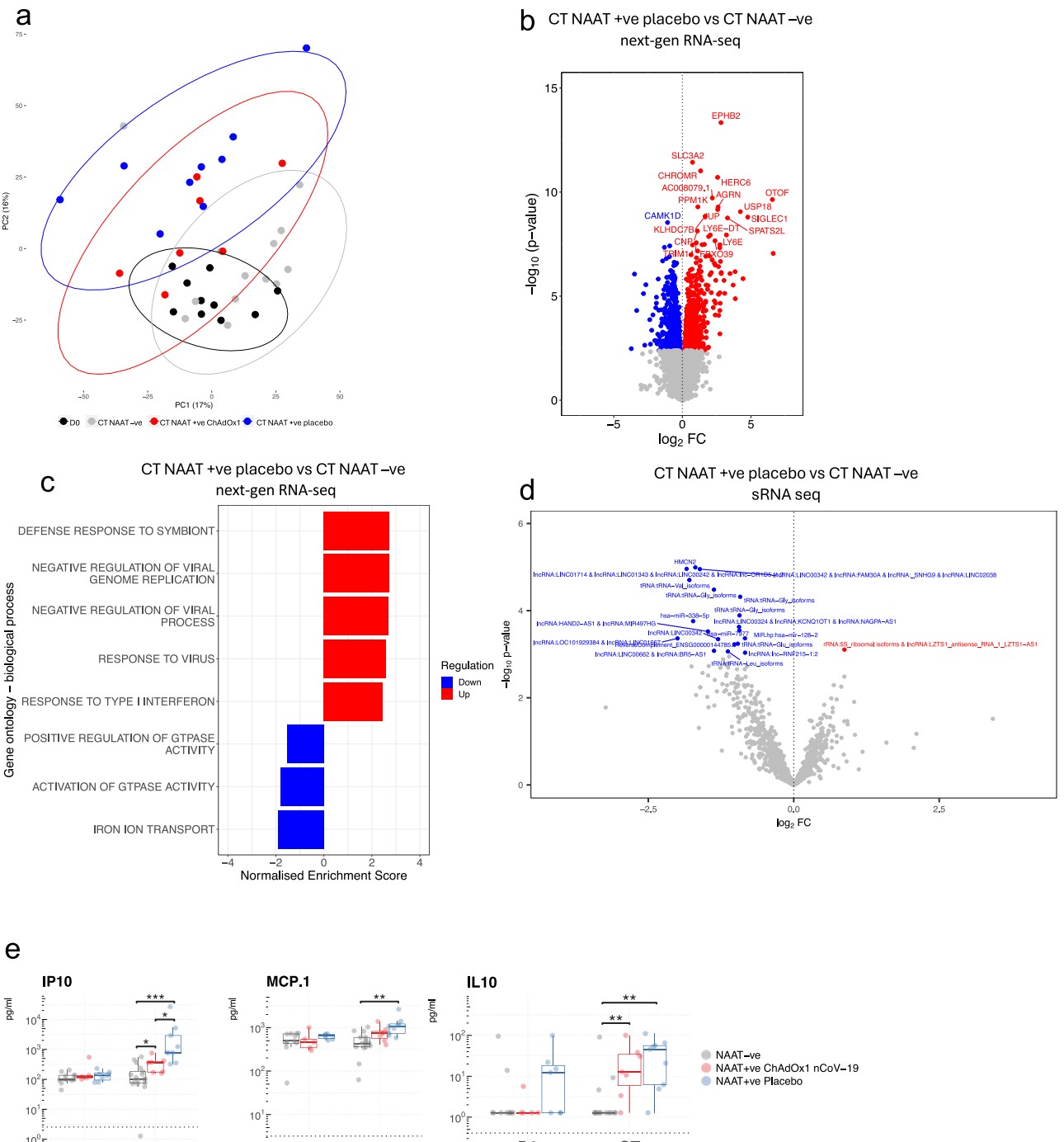

**Fig. 2 | Differential signatures and features in acute COVID-19 compared with symptomatic controls. a** Principal component analysis of blood RNA-seq transcriptome (next-gen RNA-seq) of study participants during symptomatic episodes consistent with COVID-19, with 95% confidence intervals ellipses. D0 n = 10, CT NAAT-ve n = 13, CT NAAT+ve ChAdOx1 nCoV-19 n = 7, CT NAAT+ve placebo n = 9. **b** Volcano plot comparing the next-gen RNA-seq blood transcriptome of NAAT+ve (placebo vaccine, n = 9) and NAAT−ve individuals at CT (n = 13). Blue = higher in NAAT-ve (FDR < 0.05), red = higher in NAAT+ve (FDR < 0.05). Differential expression analysis was performed using a two-sided moderate *t* test. **c** GSEA gene ontology biological process when comparing NAAT+ve (placebo vaccine, n = 9) compared with NAAT-ve individuals (n = 13) at CT. **d** Volcano plot comparing the small RNA-seq blood transcriptome of NAAT+ve (placebo vaccine, n = 9) and symptomatic NAAT-ve individuals (n = 13) at CT. Blue = higher in at NAAT-ve

(FDR < 0.05), red = higher in NAAT+ve (FDR < 0.05). Differential expression analysis was performed using a two-sided moderate *t* test. **e** Serum cytokine levels measured at baseline (D0, paired samples from NAAT-ve n = 11, NAAT+ve ChAdOx1 nCoV−19 n = 5, NAAT+ve placebo n = 7) and CT (NAAT-ve n = 17, NAAT+ve ChAdOx1 nCoV-19 n = 7, NAAT+ve placebo n = 9). Each dot represents a volunteer. The centre line denotes the median value (50th percentile, Q2), the box contains the 25th (Q1) to 75th (Q3) percentiles of dataset. The whiskers mark the Q1 – 1.5*IQR and Q3 + 1.5*IQR. Dashed line represents limit of detection. Values below the limit of detection were assigned a value of half the limit of detection. P values were derived from a two-sided unpaired Wilcoxon tests. Multiple testing correction was applied using the Benjamini-Hochberg method with an FDR < 0.05 deemed significant. Source data are provided as a Source Data file.

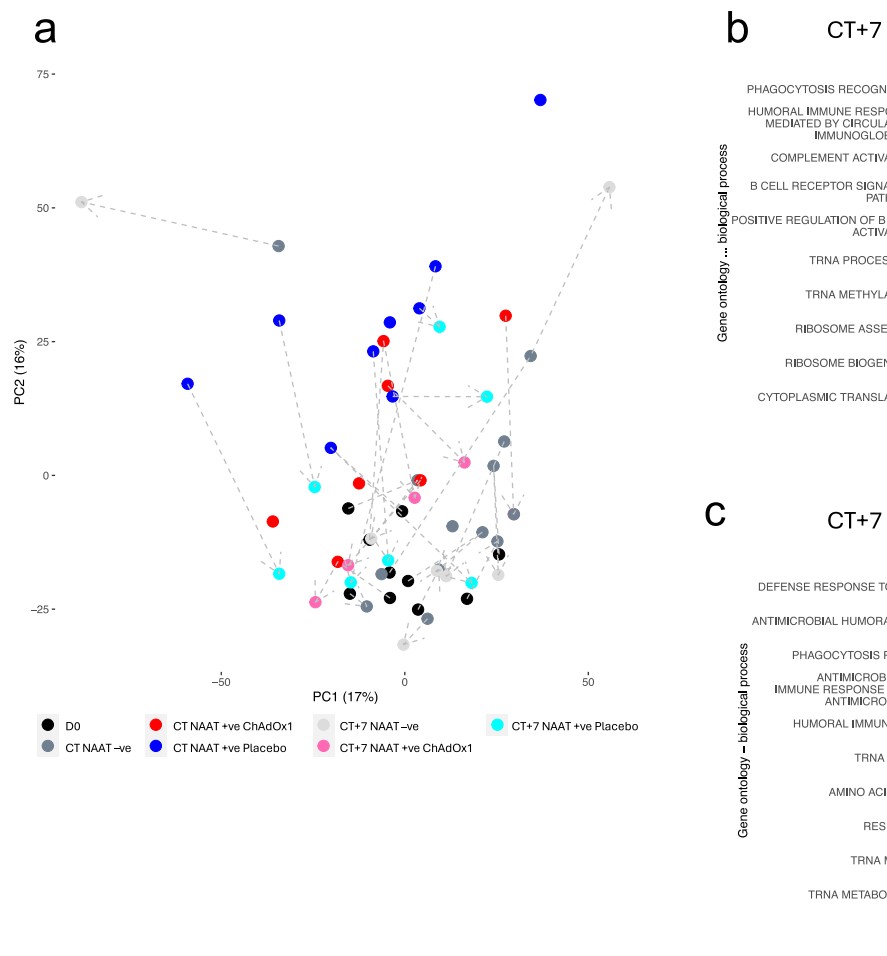

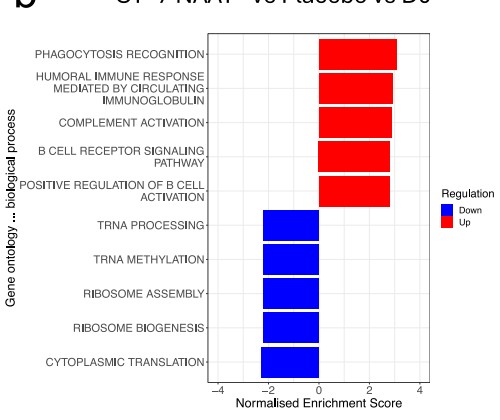

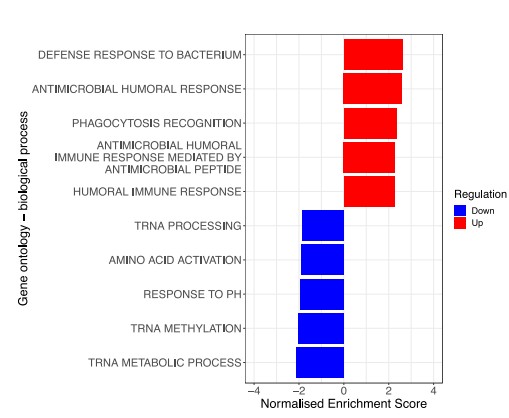

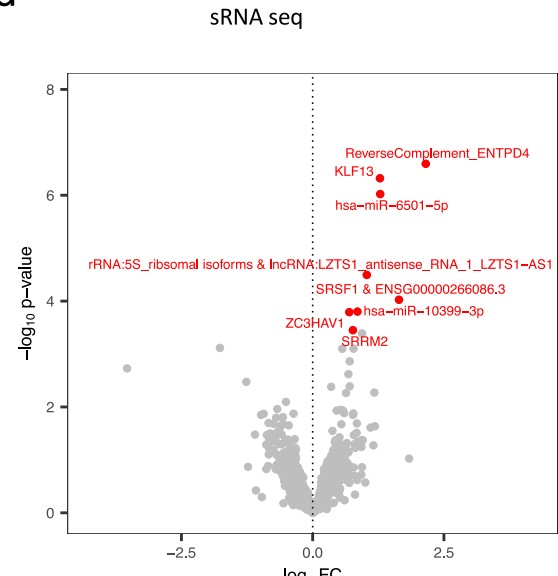

placebo and ChAdOx1 nCoV-19 vaccinated groups − no statistically significant difference was seen (Fig. 6f, Wilcoxon test, p = 0.79 for stage 1, p = 0.93 for stage 2; p = 0.93 overall). In addition, C-reactive protein (CRP) levels were similar during COVID-19 infection between the vaccine groups; median CRP in the placebo group was 3.7 (IQR 2.5,5.5), median CRP in ChAdOx1 nCoV-19 group was 3.85 (IQR 2,8.6), Mann Whitney p = 0.77 (Supplementary Fig. 9).

## ChAdOx1 nCoV-19 induces adaptive immune responses at COVID onset

ChAdOx1 nCoV-19-specific responses were sought by identifying molecular features exclusively differentially upregulated at CT in the ChAdOx1 nCoV-19 vaccine group. While no such genes were identified consistently across both study stages, GSEA on the stage 2 data revealed several B cell-related pathways and simulated CD4 T cell

**Fig. 3 | Signatures of COVID-19 response seven days after symptoms onset.**
**a** Principal component analysis of blood RNA-seq transcriptome (next-gen RNA seq) of study participants before and around symptomatic episode consistent with COVID-19. The dashed arrows connect the different time points from the same participants; the direction of the arrows is pointing towards the latter time point. D0 n = 10, CT NAAT-ve n = 13, CT NAAT+ve ChAdOx1 nCoV-19 n = 7, CT NAAT+ve placebo n = 9, CT + 7 NAAT-ve n = 7, CT + 7 NAAT+ve ChAdOx1 nCoV-19 n = 5, CT + 7 NAAT+ve placebo n = 7. **b** GSEA gene ontology biological process at CT + 7 in NAAT +ve placebo vaccinated individuals (n = 7) compared with baseline placebo samples (n = 10). **c** GSEA gene ontology biological process at CT + 7 NAAT+ve (ChAdOx1 nCoV-19) individuals (n = 5) compared with baseline (D0) samples (n = 10). **d** Volcano plot comparing the small RNA-seq blood transcriptome at CT + 7 in NAAT+ve placebo vaccinated individuals (n = 7) compared with baseline (D0) samples (n = 10). Blue = downregulated at CT + 7 (FDR < 0.05), red = upregulated at CT + 7 (FDR < 0.05). The top 20 differentially expressed genes − ranked by FDR− are labelled, with the omission of overlapping labels. Differential expression analysis was performed using a two-sided moderate t-test. Source data are provided as a Source Data file.

pathways that were exclusively enriched in ChAdOx1 nCoV-19 vaccinees at onset of COVID-19 (Fig. 6d). ChAdOx1 nCoV-19 specific responses were also seen in the sRNA data; a global upregulation of snoRNA-derived sRNAs occurred at CT in the ChAdOx1 nCoV-19 group was not seen until CT + 7 in the placebo arm (Fig. 7a), see Supplementary Fig. 17 for the relative abundance of each sRNA class. Though no miRNAs were unilaterally differentially expressed in the ChAdOx1 nCoV-19 group during COVID-19, nCov-19-specific differences were observed at the pathway level. For example, by CT + 7, the GO pathway "negative regulation of interleukin-1-mediated signalling pathway" was positively enriched in the placebo group compared with the ChAdOx1 nCoV-19 group (Supplementary Fig. 19). This pathway was positively enriched at CT + 7 compared with baseline in the placebo group (p = 0.016) but not the ChAdOx1 nCoV-19 group (p = 0.07). This suggesting increased miRNA repression of this pathway in the placebo vaccinees at CT + 7 compared with the ChAdOx1 nCoV-19 group. Increased repression of mRNAs involved in "negative regulation of interleukin-1-mediated signalling pathway" implies potential increased IL-1 signalling at CT + 7 in the placebo group 7 days into infection compared with ChAdOx1 nCoV-19 vaccinees.

## COVID-19 causes a decrease in platelets and all leucocyte subsets

CIBERSORTx deconvolution was used to estimate changes in cell abundances in blood samples taken during COVID-19 compared with baseline. This suggested a relative drop in neutrophil abundance at the onset of COVID-19 symptoms in the Placebo vaccinated individuals compared with the ChAdOx1 nCoV-19 vaccinees (Fig. 6i). This finding was corroborated by the absolute neutrophil count (ANC) data: ANCs decreased at COVID-19 onset in both study groups but were significantly lower in the placebo group, with 16% of these participants developing clinical neutropenia (ANC < $1.5 \times 10^9$ /litre) (Fig. 6j). This finding was corroborated by the absolute neutrophil count (ANC) data generated in the full blood counts taken in the full cohort of people on the ChAdOx1 nCoV-19 COVID RCTs (n = 1353) (Fig. 6i, j)[6]. COVID-19 also reduced lymphocyte, monocyte, basophil, eosinophil and platelet counts in both vaccine groups but significantly more so in placebo vaccinees (Supplementary fig. 14).

## Putative alternative SARS-CoV-2 receptor downregulated during COVID-19 infection in Placebo vaccinated individuals compared with ChAdOx1 nCoV-19 recipients

Genes differentially expressed between vaccine groups during COVID-19 infection included downregulation of *KREMEN1* in placebo vaccinees (Fig. 6g). These results were consistent in both stage 1 and stage 2 cohorts (Supplementary fig. 7). This result was further pursued as KREMEN1 has been suggested as an alternative receptor for SARS-CoV-2 entry into host cells[22]. To investigate which cells may account for downregulation in *KREMEN1*, publicly available single-cell RNA-sequencing data was analysed that showed expression of *KREMEN1* is mainly restricted to leucocytes, including neutrophils, monocytes, T cells and B cells[23] (Fig. 6h). Whilst a drop in absolute neutrophil count was observed, cell-specific expression imputed via CIBERSORTx predicted that KREMEN1 was downregulated in the neutrophils of Placebo vaccinated individuals at COVID-19 onset (Supplementary fig. 8). This suggests that the reduction

in *KREMEN1* observed is due to decreased cellular expression and not solely to a drop in neutrophils during COVID-19.

## Release of miRNA-mediated post-transcriptional repression may promote an inflammatory response in COVID-19

Differential gene expression analysis in the stage 2 cohort revealed global downregulation of individual miRNAs at COVID-19 onset (Fig. 7a). The widespread release of miRNA-mediated post-transcriptional repression coincided with upregulation of pathways involved in anti-viral and inflammatory responses. Conversely, restoration of miRNA expression 7 days after COVID-19 onset coincided with resolution of gene perturbation (Fig. 7a), with miRNAs that target pathways involved in immune responses recovering more quickly (Supplementary Fig. 15, Supplementary Data 7). This suggests that downregulation of miRNAs could play a role in promoting inflammation in COVID-19. In line with this, gene ontology pathways relating to the immune response were negatively enriched at CT compared with baseline, e.g., TRIF-dependent toll-like receptor signalling pathway, negative regulation of I-kappa B kinase, NF-kappaB signalling, positive regulation of lymphocyte activation and cell stress responses including cellular response to hypoxia and regulation of JUN kinase activity (Supplementary Fig. 11, Supplementary Data 7). Furthermore, there was a reciprocal relationship between the expression of miRNAs and their target mRNAs/pathways (Fig. 7b, c). Figure 7b shows that targets of downregulated miRNAs were enriched in the mRNA results at CT versus baseline. Figure 7c shows pathways that were positively enriched at the miRNA level were generally negatively enriched at the mRNA level and vice versa (proportion test p = $2.2 \times 10^{-15}$). These findings were consistent for CT + 7 vs baseline (Supplementary Fig. 16). In contrast to the resolving perturbation in miRNAs, there was persistence of differential expression in non-miRNA sRNAs at CT + 7 (e.g., sRNA features derived from snoRNAs). Anti-correlations were also seen at the individual miRNA-mRNA level – for example the expression of hsa-miR-150-5p and STAT1 (experimentally proven miRNA-mRNA target pairs) were anticorrelated at the sample level (Supplementary fig. 23).

## Multi-omics integration and the Coronavirus Disease COVID-19 KEGG pathway converge on TNF alpha and IFN-gamma

Many genes in the "Coronavirus Disease COVID-19" KEGG[24–26] pathway were differentially expressed at the mRNA or protein level and/or were targets of miRNAs (Supplementary Fig. 12).

The chord plot in Fig. 7d shows the overlap between differentially expressed serum cytokines, the differentially expressed RNAs that encode them and the miRNAs that target their mRNAs. Three differentially expressed proteins (TNF-alpha, IL18 and CXCL10/IP10) were detected at the mRNA level in whole blood−and of these, only IL18 and CXLC10 were differentially expressed at the mRNA level. At the pathway level, "response to TNF-alpha (GO003462)" is positively enriched in the mRNA data (p = 0.022) and negatively enriched in the miRNA data (p = 0.01). IFN-gamma was upregulated at CT and targeted by several downregulated miRNAs. Though IFN-gamma was not detected in the mRNA data, pathways relating to interferon were positively enriched in the mRNA gene set expression analyses and negatively enriched in the miRNA pathway analyses−for example, interferon-

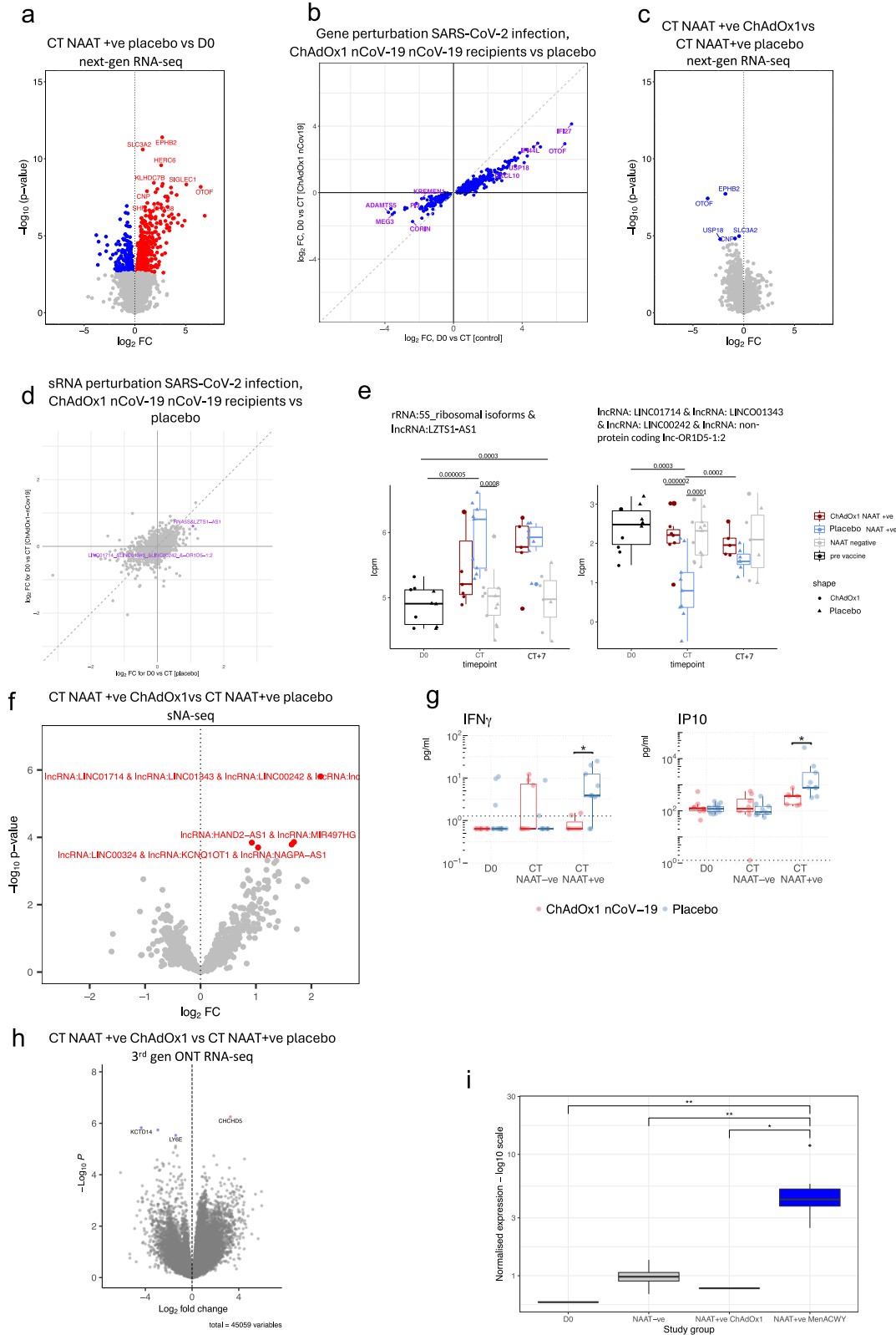

gamma-mediated signalling pathway GO0060333 was enriched in the mRNA and miRNA results ($p = 2.03 \times 10^{-9}$, $p = 5.44 \times 10^{-4}$, respectively).

## Discussion

Here, we comprehensively evaluated COVID-19 infection in the context of vaccination using a systems immunology approach. By leveraging the clinical trial setting where participants were randomised to receive either a ChAdOx1 nCoV-19 or a placebo vaccine, we captured vaccine-induced modification of response to COVID-19 infection. ChAdOx1 nCoV-19 vaccination attenuated the activation of immune responses associated with COVID-19 severity. Moreover, we described the downregulation of a putative alternative SARS-CoV-2 receptor in blood during COVID-19 infection in unvaccinated individuals compared with ChAdOx1 nCoV-19 recipients.

**Fig. 4 | Attenuated response to SARS-CoV-2 infection in ChAdOx1 nCoV-19 vaccinees. a** Volcano plot comparing next-gen RNA-seq blood transcriptome at CT in NAAT+ve (placebo vaccine) with baseline samples. Blue = downregulated at CT, red = upregulated at CT. **b** Agreement plot of differentially expressed genes at CT in NAAT+ve individuals in Placebo (x-axis) and ChAdOx1 nCoV-19 (y-axis) vaccinees compared with baseline. **c** Volcano plot comparing the next-gen RNA-seq blood transcriptome at COVID-19 onset in ChAdOx1 nCoV-19 vaccinees compared with placebo vaccinees. **d** Agreement plot of small RNA-seq differentially expressed genes at CT in NAAT+ve individuals in Placebo (x-axis) and ChAdOx1 nCoV-19 (y-axis) vaccinees compared with baseline. Coloured features differentially expressed in placebo arm (FDR < 0.05). **e** Boxplots of sRNA features differentially expressed (FDR < 0.05) between baseline and CT in the placebo vaccine group. Y-axis shows unadjusted p values. **f** Volcano plot comparing the small RNA-seq blood transcriptome at COVID-19 onset (NAAT +ve CT) in ChAdOx1 nCoV-19 vaccinees compared with placebo vaccinees. Red = higher in placebo group. **g** Serum cytokine levels measured at D0 (paired samples NAAT+ve ChAdOx1

nCoV-19 n = 5, NAAT+ve placebo n = 7) and symptom onset in Placebo (NAAT-ve n = 9, NAAT+ve n = 9) and ChAdOx1 nCoV-19 (NAAT-ve n = 8, NAAT+ve n = 7) vaccinees. Dashed line represents limit of detection. P values were derived from two-sided unpaired Wilcoxon tests. **h** Volcano plot comparing the 3rd gen ONT RNA-seq blood transcriptome COVID-19 onset (NAAT+ve CT) in ChAdOx1 nCoV-19 vaccinated participants compared with placebo vaccinees; 45,231 transcripts included in total. **i** MX1 ENST00000288383 transcript (MX1-201 isoform) expression across the study groups. Group sizes: D0 n = 10, CT NAAT-ve n = 13, CT NAAT+ve ChAdOx1 nCoV-19 n = 7, CT NAAT+ve placebo n = 9. Statistics: Unless otherwise stated above, differential gene expression analyses were performed using a two-sided moderated *t* test. Multiple testing correction applied to all differential expression analyses using the Benjamini-Hochberg method with an FDR < 0.05 deemed significant; *=FDR < 0.05, **=FDR < 0.01. Box plots: the centre line denotes the median value, the box contains the 25th (Q1) to 75th (Q3) percentiles of dataset. Whiskers mark the Q1 − 1.5*IQR and Q3 + 1.5*IQR. Source data are provided as a Source Data file.

We found distinct blood transcriptome profiles between COVID-19 cases, non-COVID-19 symptomatic episodes, and health. In line with previous reports, we described upregulation of genes involved in response to virus and interferon signalling in blood samples from COVID-19 patients compared with healthy controls[17]. Pertinently, few studies have used another disease comparator group. Our symptomatic placebo group represent an unknown and likely heterogeneous aetiology group with commonality in COVID-19-like symptomology, allowing us to identify SARS-CoV-2-associated immune responses distinct from general immune responses to infection[27].

This study shows that vaccination ameliorates molecular perturbation in blood during COVID-19, a finding corroborated across all omics datasets. A reduction in viral load in the upper respiratory tract did not explain these effects. Although our cohort may be underpowered to observe small differences between the vaccine groups, larger studies also show a poor relationship between vaccine status and viral load[12,13]. Notably, previously described biomarkers of severe COVID-19, namely IP10, IL18 and IFN-gamma, were more highly expressed in the placebo vaccine group in our study at the protein level in serum[28]. Moreover, IL-10, IL6 and TNF-alpha, also biomarkers of COVID-19 severity[29,30], were exclusively upregulated in the placebo group at the protein level in serum. Transcriptomic responses mirrored this finding, with enrichment of pathways involved in type I interferons at CT in Placebo compared with ChAdOx1 nCoV-19 vaccinees.

Interestingly, we described a truncated isoform of MX1 protein, MX1-201, upregulated at the onset of COVID-19 in placebo group participants compared with ChAdOx1 nCoV-19 vaccinees. Human myxovirus resistance protein 1 (MX1 or MxA) is an interferon-induced GTP-binding protein encoded by the MX1 gene, which plays an important role in anti-viral activity by inhibiting a range of viruses, including influenza, parainfluenza, measles, coxsackie, and hepatitis B virus[31]. Since the truncation significantly alters the protein structure, we hypothesise that it could potentially destabilise the hydrophobic core of the stalk and impair the oligomerisation of MX1, thereby making this isoform dysfunctional. The expression of the MX1-201 transcript may alter the immune response to the COVID-19 infection in the placebo vaccine recipients compared with ChAdOx1 nCoV-19 vaccinated group, where this isoform was almost absent. Interferon pathways have been associated with disease severity[32]. However, results for the link between type 1 interferon and COVID-19 severity are conflicting and may vary over COVID-19 time course[32–37]. One hypothesis is that elevated type 1 interferon responses drive TNF- and IL-1-associated inflammation[32]; thus, vaccine-associated attenuation of type I interferon may explain how ChAdOx1 nCoV-19 reduces disease severity.

Given that ChAdOx1 nCoV-19 induces both spike-specific CD8 + T-cells and antibodies[19,38], our study ties adaptive immune responses

against spike to reduced stimulation of pro-inflammatory responses associated with immunopathology in SARS-CoV-2 infection. Supporting this, we found B cell and CD4 T cell pathways enriched in ChAdOx1 nCoV-19 vaccinees compared with controls at COVID-19 onset, revealing rapid, vaccine-induced anti-SARS-CoV-2 adaptive immune responses captured in real-time. Notably, the study revealed that vaccination attenuates pro-inflammatory responses rather than reducing viral load, decoupling control of infection from disease severity. Future studies focusing on mucosal early responses to infection in the context of vaccination could inform whether vaccine-induced decreased pro-inflammatory responses in the mucosa are associated with reduced viral load and/or faster viral clearance and thus potentially reduced transmission. Understanding this biology will be crucial for second-generation transmission-blocking vaccines.

Our findings showed that COVID-19 did not induce Th2-skewed immune responses in ChAdOx1 nCoV-19 vaccinated individuals. This is believed to be a desirable attribute for vaccines against respiratory viruses as some studies proposed that Th2-skewed immune responses are the basis for immunopathology in historical examples of vaccine-enhanced disease in animal models and humans[28,39–43]. The ability of viral vector vaccines (such as ChAdOx1) to induce Th1 responses was known from preclinical and clinical studies and informed the design of this vaccine.

We measured the downregulation of *KREMEN1* at the onset of COVID-19 in placebo vaccinated individuals compared with ChAdOx1 nCoV-19 recipients. Interestingly, Kringle containing transmembrane protein 1 (KREMEN1) is a transmembrane receptor involved in WNT signalling that has recently been proposed as an alternative receptor for the SARS-CoV-2 virus[22]. KREMEN1 is a known entry receptor for enteroviruses, such as coxsackievirus A10, and is expressed at potential virus entry sites (i.e., sinonasal, conjunctival and bronchiolar epithelium)[44,45]. In blood, *KREMEN1* expression is mainly restricted to leucocytes, including neutrophils, monocytes, T cells and B cells. While we observed a drop in absolute neutrophil count, we also predicted a cell-specific decrease in *KREMEN1* in the neutrophils of Placebo vaccinated individuals at COVID-19 onset compared with ChAdOx1 nCoV-19 vaccinees. SARS-CoV-2 has been demonstrated to infect and replicate within human neutrophils[46]. This has been shown to lead to the activation of neutrophils —including triggering of neutrophil extracellular traps — a process that can be inhibited by blocking interactions of S protein within the ACE2 receptor[46,47]. This raises the possibility that the expression of viral entry receptors could impact the infection and activation of neutrophils, subsequently impacting their migration and/or survival.

Early in COVID-19 infection, we observed a mild but statistically significant drop in blood counts across all leucocyte subsets and platelets, which was more profound in placebo vaccinees. In a COVID-19 challenge study, neutropenia was a common finding; however,

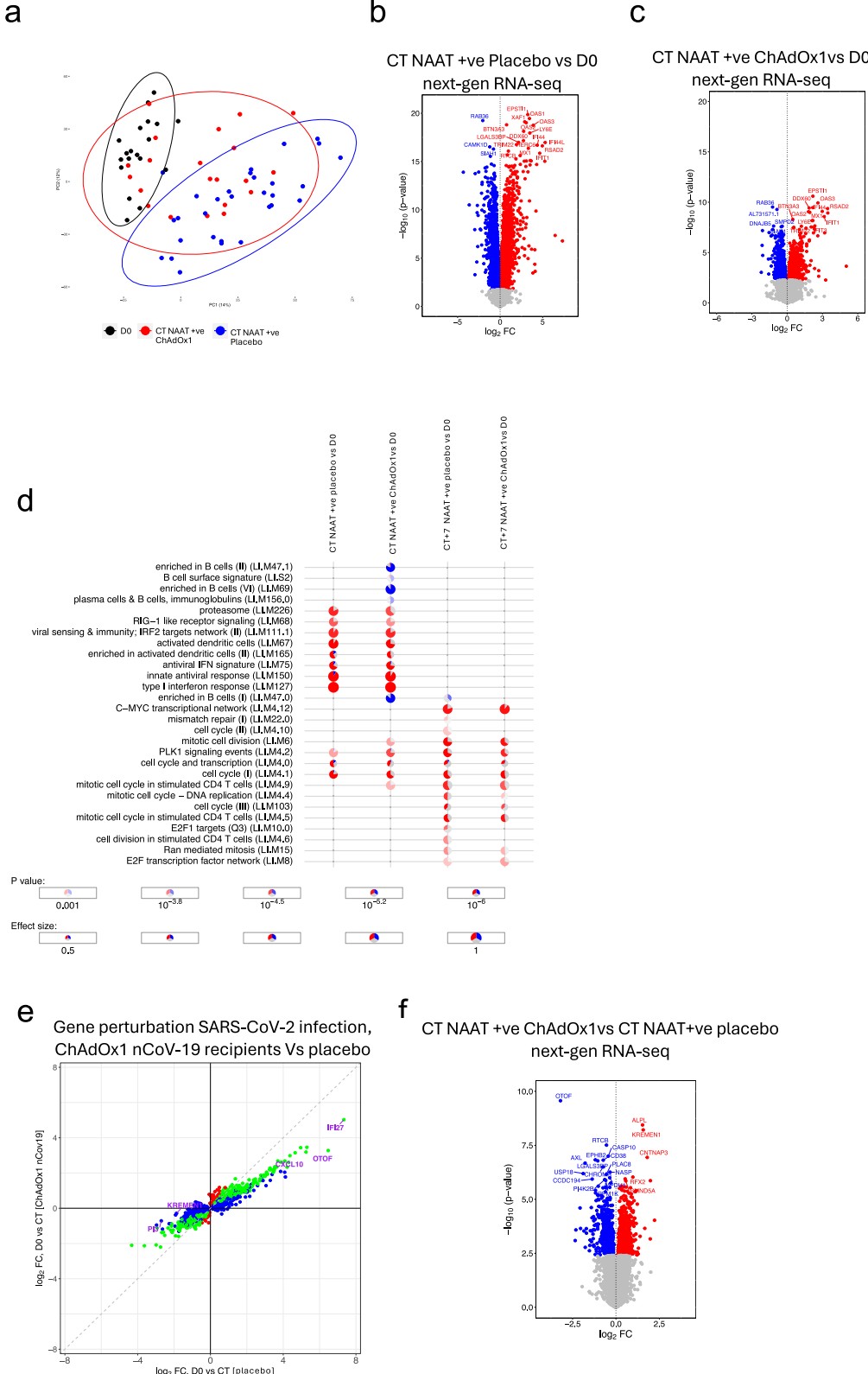

e **Gene perturbation SARS-CoV-2 infection, ChAdOx1 nCoV-19 recipients Vs placebo**

f **CT NAAT +ve ChAdOx1vs CT NAAT+ve placebo next-gen RNA-seq**

changes in other cell types were not described[48]. Most studies recruit later in infection and contain more severe cases, potentially explaining why this has not been reported previously. A potential explanation for the observed fall in all leucocyte populations and platelets is transient bone marrow suppression. However, other possibilities exist, such as cell migration out of circulation and into infected tissues or destruction within peripheral blood, previously described in viral

infections[49,50]. The decrease in granulocyte counts during COVID-19 was not associated with changes in levels of GCSF or GM-CSF in our cohort, composed mainly of mild illness. However, the drop in blood leucocyte counts may explain the emergency myelopoiesis and increase in immature neutrophils frequently observed in severe or critical COVID-19[17]. Despite our mostly mild cases, we noted markers of immature neutrophils suggesting this phenomenon is pervasive in

**Fig. 5 | Signatures of COVID-19 response from transcriptomics in larger cohort of NAAT+ participants. a** Principal component analysis of blood RNA-seq transcriptome (next-gen RNA-seq) of study participants before (D0, n = 19) and at the time of COVID-19 (CT NAAT+ve ChAdOx1 nCoV-19 n = 21, CT NAAT+ve placebo n = 30). **b** Volcano plot comparing the next-gen RNA-seq blood transcriptome at COVID-19 onset (CT NAAT+ve), placebo vaccinees (n = 30) compared with baseline (D0, n = 19) samples. Blue = downregulated at CT (FDR < 0.05), red = upregulated at CT (FDR < 0.05). Differential expression analysis was performed using a two-sided moderate *t* test. **c** Volcano plot comparing the next-gen RNA-seq blood transcriptome at COVID-19 onset (CT NAAT+ve) in ChAdOx1 nCoV-19 vaccinees (n = 21) compared with baseline (D0, n = 19) samples. Blue = downregulated at CT (FDR < 0.05), red = upregulated at CT(FDR < 0.05). Differential expression analysis was performed using a two-sided moderate *t* test. **d** Blood transcriptional modules

(BTM) enriched during COVID-19 compared with baseline. Enriched BTMs (FDR < 0.001) are displayed. Segments of the pie charts represent the proportion of upregulated (red) and downregulated (blue) genes (absolute fold change >1.25). Enrichment P values were derived from a hypergeometric test, after adjustment for multiple testing (Benjamini and Hochberg's method). **e** Agreement plot of differentially expressed genes at CT NAAT+ve in individuals in Placebo (x-axis, n = 30) and ChAdOx1 nCoV-19 (y-axis, n = 21) vaccine recipients compared with baseline (D0, n = 19). **f** Volcano plot comparing the next-gen RNA-seq blood transcriptome at CT NAAT+ve in individuals who received the ChAdOx1 nCoV-19 vaccine (n = 21) compared with individuals who received a placebo vaccine (n = 30). Blue = higher in placebo group (FDR < 0.05), red = higher in ChAdOx1 nCoV-19 group (FDR < 0.05). Differential expression analysis was performed using a two-sided moderate *t* test. Source data are provided as a Source Data file.

COVID-19 infection, starting early after onset and abrogated by the ChAdOx1 nCoV-19 vaccination.

Our study shows COVID-19 perturbs whole blood sRNA expression, with an overlap noted between differentially expressed miRNAs in this study and other studies[51–59]. For example, the global reduction in miRNA expression at COVID-19 onset was observed in a study of plasma miRNAs in patients with moderate to severe COVID-19[59]. The same phenomenon has been linked to greater COVID-19 disease severity based on miRNA measurement in nasopharangeal swab samples[60].

The mechanism of global miRNA downregulation at COVID-19 onset is unclear. At the gene expression level, DICER and three AGO proteins, important for miRNA maturation and function, respectively[61–63], were down-regulated—a phenomenon known to globally reduce miRNAs[64,65]. Another possibility is increased miRNA turnover, given RBX1, part of the ZSWIM8 complex involved in target-directed miRNA decay, was upregulated at the mRNA level[66]. The link between global miRNA downregulation and de-repression of mRNAs and pathways, including those involved in pro-inflammatory responses, potentially implicates dysregulated miRNA expression in COVID-19 immune responses. This fits with literature showing that miRNA downregulation is associated with inflammatory states[65]. Widescale loss of miRNA gene regulation was specific to COVID-19, as miRNA pathway analyses at symptom onset identified increased miRNA-mediated pathway repression in the symptomatic COVID-19 negative compared with the COVID-19 group. This potentially reveals a distinct aspect of COVID-19 pathology. Overall, these results raise the possibility that dysregulated miRNA expression contributes to the pathogenic inflammatory response that makes COVID-19 such a serious disease. Functional work is required to prove causation between miRNA expression and inflammation in COVID-19, nevertheless such a possibility is supported by multiple studies showing that global miRNA reduction is associated with increased cytokine production and excessive inflammation[67–70]. Our results open the door to further research on whether reinstating global or specific miRNA levels could ameliorate COVID-19 severity.

The changes seen in snoRNA-derived sRNA are supported by the findings of Wilson et al., who described this phenomenon later in COVID-19[59]. Notably, sRNAs derived from snoRNAs were higher in ChAdOx1 nCoV-19 vaccinees at symptom onset, with a delayed increase appearing at CT + 7 in placebo vaccinees. sRNAs derived from snoRNAs are believed to act like miRNAs, post-transcriptionally regulating gene expression[48,56,57] and can be differentially expressed during inflammatory diseases[71,72]. The temporal difference in the expression of snoRNA-derived sRNAs between the ChAdOx1 nCoV-19 and placebo vaccinees fits with the time course for the emergence of adaptive immunity in the two groups[59]. This implies that snoRNA-derived sRNAs play a role in adaptive immune responses, a phenomenon not previously described.

In conclusion, our study shows that vaccination with ChAdOx1 nCoV-19 ameliorates the molecular response to SARS-CoV-2 infection.

Concurrent with this is the early emergence of pathways involved in humoral and cellular immunity in ChAdOx1 nCoV-19 vaccinees, tying a reduction in disease severity with adaptive immune responses at the molecular level. These data reveal immune-related biomarkers that can be used to identify vaccine benefit beyond sterilising immunity in a cohort of people generally at low risk of COVID-19. Our multi-omics approaches provide corroborative evidence that labelling breakthrough infection as "vaccine failure" is a misnomer, as there is clear evidence of impact at the molecular level. Our study also adds to the mechanistic understanding of COVID-19 immune responses, with a link between pro-inflammatory pathways at COVID-19 onset and a reduction in miRNA-mediated repression. Our results provide a mechanistic understanding of real-world data showing how ChAdOx1 nCoV-19 reduces disease severity. This is relevant to the emerging paradigm shift in vaccinology that refocuses vaccine evaluation away from sterilising immunity towards disease attenuation for pathogens where sterilising immunity may not be necessary or achievable.

## Methods

### Samples and study design

The "COV001" and "COV002" studies were observer-blinded, randomised controlled trials (RCTs) designed to determine the safety and efficacy of the Oxford-AstraZeneca COVID19 vaccine ChAdOx1 nCoV-19, as previously described[6,19]. The RCTs are registered at ClinicalTrials.gov, NCT04324606 (COV001) and NCT04400838 (COV002). All individuals included in the analyses agreed to inclusion in this analysis as part of the vaccine trial consent, with the opportunity to opt-out. The trials were conducted according to the principles of Good Clinical Practice and approved by the South Central Berkshire Research Ethics Committee (20/SC/0145 and 20/SC/0179) and the UK regulatory agency (the Medicines and Healthcare products Regulatory Agency). All participants included in this work had no history of laboratory-confirmed SARS-CoV-2 infection or COVID-19-like symptoms prior to enrolment. Participants were randomised to receive ChAdOx1 nCoV-19 or meningococcal group A, C, W, and Y conjugate vaccine (MenACWY) and blinded to their allocation. The original protocol involved a single dose schedule, however an amendment was made to offer a booster dose (ChAdOx1 nCoV-19 / placebo) was implemented from 3rd August 2020. Participants second vaccination was the same as their initial vaccine. More information is contained in the original COV1/COV2 study paper[6]. Participants presented for a COVID-19 test visit (CT) during the study period as soon as possible after the onset of symptoms of COVID-19 (cough, shortness of breath, fever, anosmia or ageusia). At this visit, they were medically assessed, a COVID-19 nucleic acid amplification test (NAAT) was performed, and blood was taken for transcriptomic analysis or processed for serum cytokine assays. Seven days later, participants reattended for the same procedures (CT + 7). Participants then returned to standard follow-up until their next episode of COVID-like symptoms, when they would have another CT visit. See Fig. 8 for study schema. Blood samples were collected from participants with COVID-like symptoms who did (NAAT+ve) and did not

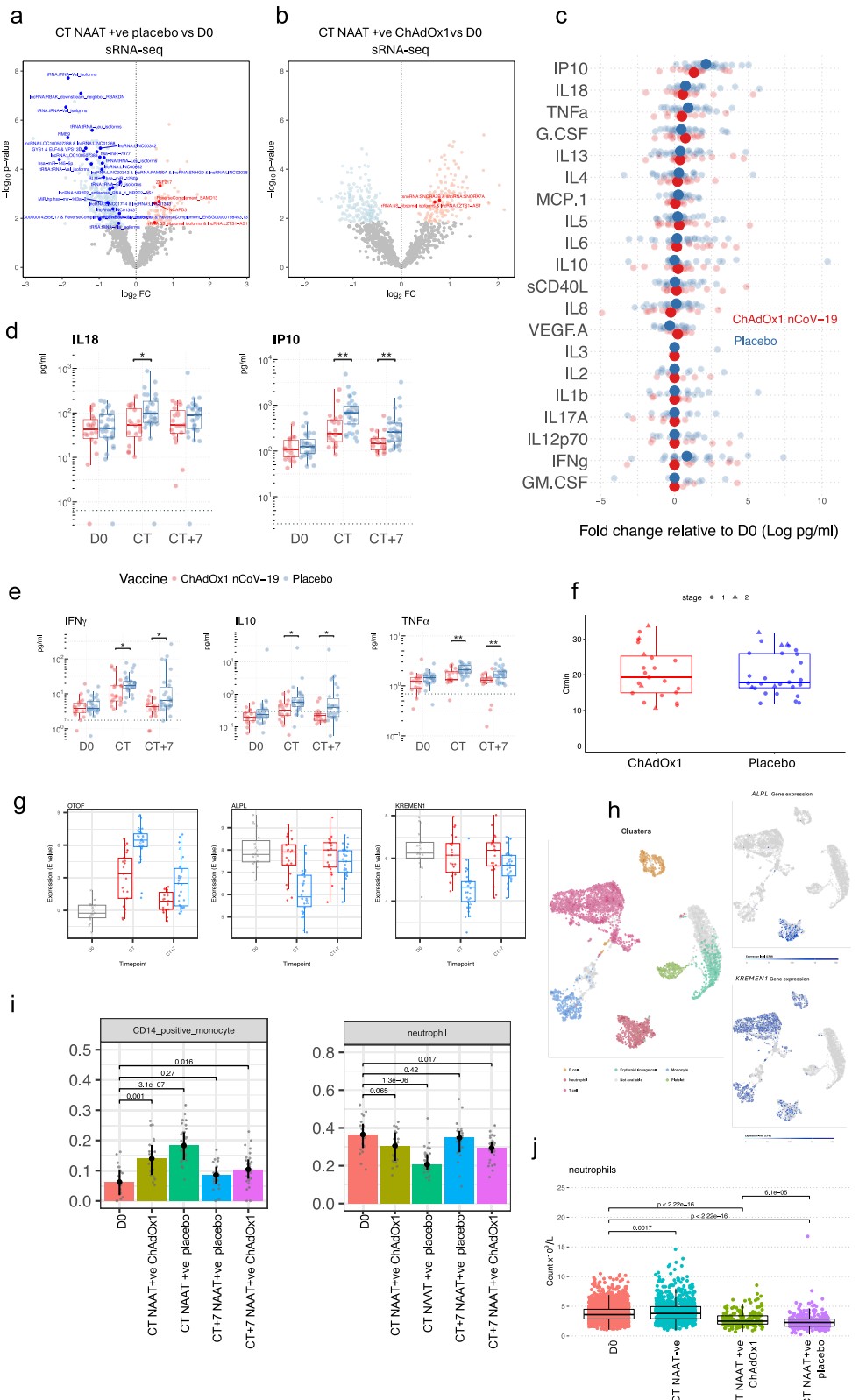

(NAAT-ve) have COVID-19. NAAT-ve episodes thus represent a heterogeneous group of COVID-like illnesses. We analysed the study in two stages. Stage 1 involved samples taken at baseline, CT and CT + 7 from people who did and did not have SARS-CoV-2 infection. Stage 2 included a larger group of participants, independent from the stage 1, and focussed on the differences between vaccine groups during acute COVID-19; thus, it did not contain samples from NAAT-ve episodes.

**Sample selection.** Baseline samples were not available for all participants. For this reason, unpaired baseline samples were selected from participants matched for important COVID-19 risk factors (age, sex, body mass index and comorbidities). Samples from ChAdOx1 nCoV-19 and placebo vaccine groups were selected to be matched for COVID-19 risk factors, as were samples from NAAT+ve and NAAT-ve individuals. Demographic matching of trial participants dictated which samples

**Fig. 6 | Dissecting attenuation of the response in vaccines.** Volcano plot comparing the small RNA-seq blood transcriptome at CT NAAT+ve (placebo, n = 30) (**a**) and at CT NAAT+ve in ChAdOx1 nCoV-19 vaccinees (n = 21) (**b**) compared with baseline (D0, n = 19) samples (blue downregulated FDR < 0.05, red upregulated FDR < 0.05, bold p < 0.05 in stage 1 data). Differential expression analysis was performed using a two-sided moderate $t$ test. Serum cytokine levels measured by Luminex (**c**, **d**) and MSD (**e**) in paired samples from NAAT+ve ChAdOx1-nCoV-19 (n = 18 D0, n = 19 CT and CT + 7, red) and NAAT+ve placebo (n = 31 D0 and CT, n = 30 CT + 7, blue). **c** Fold-change in cytokines at symptom onset relative to baseline. Solid dots represent mean fold-change, translucent dots represent individual volunteers. **d**, **e** Boxplots describing representative concentrations of cytokines. Statistical significance assessed based on FDR from two-sided unpaired Wilcoxon tests. Dashed line represents limit of detection. **f** Minimum cycle threshold (Ctmin) values of RT-PCR for the COVID-19 episode in placebo and ChAdOx1 nCoV-19 vaccine recipients, two-sided Wilcoxon test, $p = 0.79$ for stage 1 (ChAdOx1 nCoV-19 n = 5, placebo n = 6), $p = 0.93$ for stage 2 (ChAdOx1 nCoV- 19 = 18, placebo n = 24); p = 0.93 overall. **g** Top three genes differentially regulated, by p value, during COVID-19 in ChAdOx1-nCoV-19 vaccinees (red, CT n = 21, CT + 7 n = 21) compared with placebo vaccine recipients (blue, CT n = 21, CT + 7 n = 31). **h** RNA-sequencing data from single cell expression atlas (https://www.ebi.ac.uk/gxa/sc/experiments/E-MTAB-9221/) showing clusters by cell identity and expression of *ALPL* and *KREMEN1* in cell clusters. **i** Estimated cell abundance from RNA-seq data derived from CibersortX at baseline (D0, n = 19) and in NAAT+ve participants at CT (ChAdOx1 nCoV-19 n = 21, placebo n = 30) and CT + 7 (ChAdOx1 nCoV-19 n = 21, placebo n = 31). Bar denotes the median value. **j** Absolute neutrophil counts measured at baseline (n = 2276), and at CT in NAAT+ve (ChAdOx1 nCoV-19 vaccinees n = 190, placebo vaccinees n = 327) and NAAT-ve (n = 835) participants. Multiple testing applied in differential gene and protein analyses using the Benjamini-Hochberg method (*FDR < 0.05, **FDR < 0.01). Boxplot (**d**–**g**, **j**) boxes contain 25th to 75th percentiles of dataset, with central lines representing medians. Whiskers mark 1.5x the inter-quantile range (IQR). Each dot represents one volunteer. Source data are provided as a Source Data file.

were included in the study and was limited to the number of breakthrough COVID-19 infections in the trial. Only samples from one symptomatic episode were included for each participant (first COVID-19 episode in the case of NAAT+ve individuals). The severity of all suspected symptomatic COVID-19 cases was classified into four groups according to the clinical study plan of the vaccine trial. Samples in this study were not selected based on disease severity; however, all COVID-19 episodes were either mild or moderate (Supplementary Table 1, Supplementary Table 5).

### RNA sequencing

**RNA extraction.** Peripheral blood (up to 2.5 ml) was collected into a PAXgene™ RNA tube just prior to vaccination, CT, and CT + 7. Total RNA was extracted using the PAXgene blood RNA kit (PreAnalytiX, Switzerland) on the QIAsymphony SP (Qiagen) with the PAX_RNA V5 protocol. Eluted RNA was quantified using the Agilent RNA ScreenTape assay on the 4200 TapeStation System (Agilent).

**RNA sequencing - next generation (next-gen) sequencing.** RNA was ribodepleted, and globin depleted using Ribo-Zero™ Gold rRNA removal Kit (Illumina, USA). RNA was converted to cDNA; second-strand cDNA synthesis incorporated dUTP. The cDNA was end-repaired, A-tailed and adapter-ligated, and prior to amplification, samples underwent uridine digestion. The prepared libraries were size-selected, multiplexed, and quality controlled before 150 bp paired-end sequencing (NovaSeq6000). Sequencing was conducted at the Wellcome Trust Centre for Human Genetics (Oxford, UK). Sequences that mapped to human rRNA genome were excluded and the remaining sequencing data were aligned against the whole human (*Homo sapiens*) genome build GRCh38 (https://ccb.jhu.edu/software/hisat2/index.shtml), using STAR (version 2.7.3a). Gene features were counted using HTSeq (version 0.11.1), using human gene annotation general transfer format version GRCh38.92 (www.ensembl.org). Both coding and non-coding genes were retained in the analysis. Genes with low counts across most libraries were removed by only retaining genes with abundance greater than 3 counts per million in 9 or more samples. Ribosomal RNA (rRNA), sex chromosome genes, mitochondrial RNA and haemoglobin genes were excluded from downstream analysis. Total number of genes/variables retained for the downstream analysis was 13, 457 and 14, 319 for stage 1 and stage 2 respectively. Density plots of log-cpm normalized data before and after filtering are available as Supplementary fig. 22(a–d). Average power vs sample size for stage 1 cohort (n = 58 samples) was calculated using R package "ssizeRNA" and is available as Supplementary fig. 22k. Human leucocyte antigen typing of RNA-sequencing data using RNA2HLA (version 1.1) was used to check correct pairing of samples collected from the same participants[73]. We refer to this as next-gen RNA sequencing to differentiate it from the sRNA-seq and long read – 3rd gen RNA sequencing described below.

**Small RNA (sRNA) sequencing.** Small RNA libraries were prepared with the NEBNext® Small RNA Library Prep Set for Illumina (catalogue #E7330) and run on a single flow cell (target of 300 million reads) on the NextSeq platform using the 75SR configuration with the aim of achieving 5.2 million reads per sample. Reads were first aligned to the non-coding transcriptome (see supplementary methods). Non-aligned reads were then aligned to the mRNA transcriptome and antisense to the non-coding and mRNA transcriptome. Alignments were performed using bowtie with up to 2 mismatches in best strata mode and bowtie2 with 2 gaps permitted (manual filtering of alignments by CIGAR string was done to create a best-strata type output). Reads mapped to a miRNA were assigned to that miRNA regardless of whether they mapped elsewhere. All other reads were collapsed using a "gene_union" approach similar to that employed in seqCluster and mmQuant[74,75] (see supplementary methods). sRNA features were removed if they did not have an abundance of at least 5 counts per million in at least 5 samples in the stage 1 study or 20% of samples in the stage 2 study. Total number of sRNA features retained for the downstream analysis was 1,113 and 1,130 for stage 1 and stage 2 respectively. Density plots of log-cpm normalized data before and after filtering are available as Supplementary fig. 22(g–j). Average power vs sample size for stage 1 cohort (n = 58 samples) was calculated using R package "ssizeRNA" and is available as Supplementary fig. 22l.

**Long-read RNA sequencing – 3rd generation (3rd gen) sequencing.** PCR-cDNA libraries were prepared using the SQK-PCS110 PCR-cDNA-sequencing kit following the standard Oxford Nanopore Technologies (ONT) protocol[76]. Samples were barcoded (#1–8) and sequenced on PromethION. Two samples were loaded per flow cell (15 ng each), with paired samples run together and unpaired samples randomly split into pairs. The base calling and demultiplexing were performed live using original ONT software. Samples were processed following the previously described pipeline denoted as salmonminimap2[77]. In brief: reads were aligned to the reference transcriptome (version GRCh38) using minimap2 (secondary score ratio -p 0.99), and the transcript abundances were estimated from the output bam files using Salmon in alignment-based mode[78]. Low abundant transcripts with less than 3 reads across less than 5 samples, which is 50% of the smallest study group, were removed. Transcripts corresponding to ribosomal RNA (rRNA), mitochondrial RNA and haemoglobin genes were excluded from downstream analysis. Total of 45,231 transcripts were included in the downstream analysis. Density plots of log-cpm normalized data before and after filtering are available as Supplementary Fig. 22e, f.

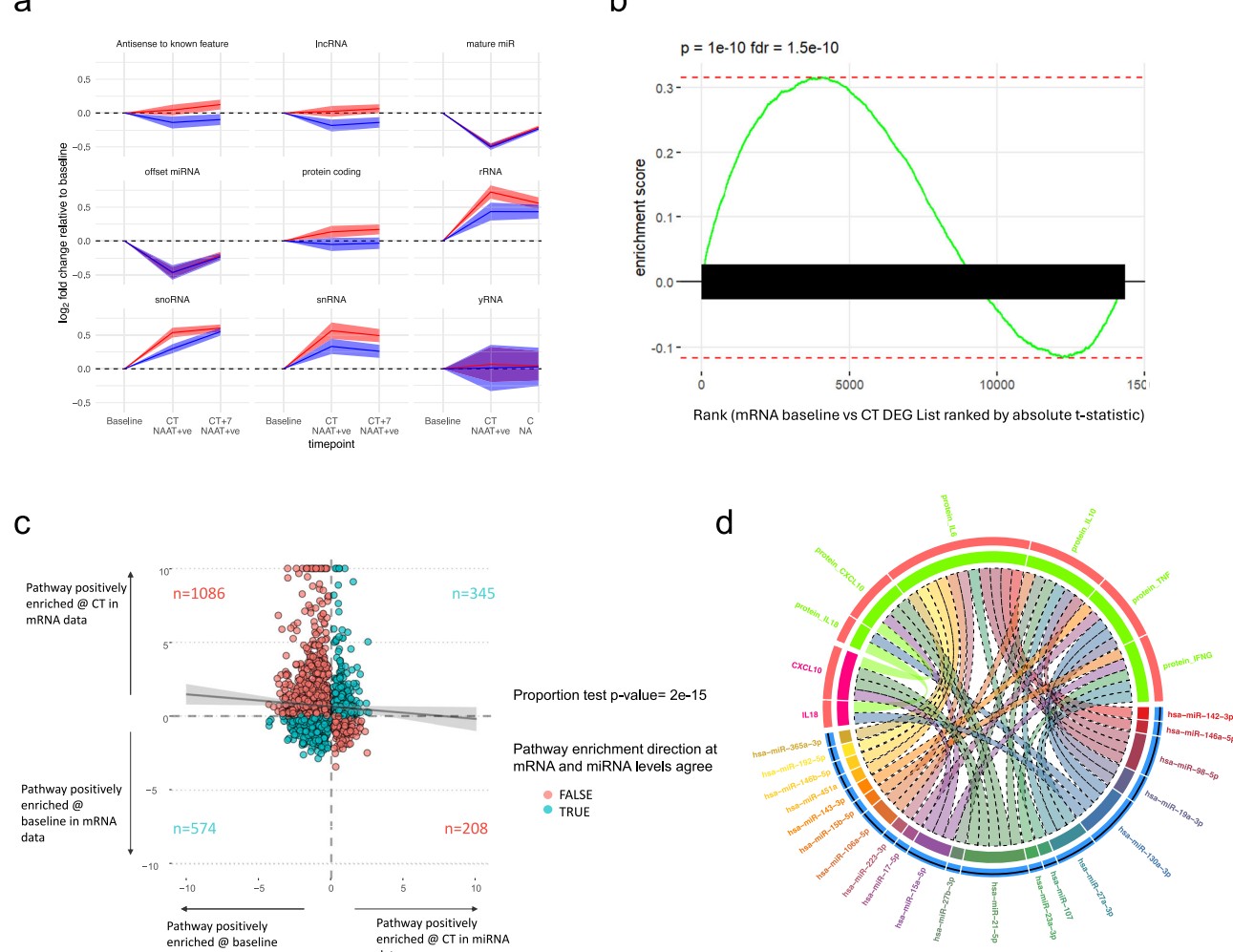

**Fig. 7 | sRNA expression and post-transcriptional regulation of COVID-19 responses. a** Class-wise changes in sRNA expression over time relative to baseline. Y-axis represents mean log2 foldchange of sRNA features within each sRNA class. 95% confidence intervals shown in shaded area. **b** Gene set enrichment result for the set of mRNAs targeted by miRNAs downregulated at CT. mRNAs ranked by t-statistic based on differential expression gene expression analysis comparing baseline with CT. P value derived from a GSEA analysis computed via the MIEA2 web-program which uses a two-sided Kolmogorow–Smirnow test and performs multiple testing correction. **c** Agreement plot for GO:BP term enrichment in the mRNA and miRNA data. Each dot is a pathway. The x-axis shows the $\log_{10}$ of the p-value of the pathway enrichment in the miRNA data. The y-axis shows the $\log_{10}$ of the p-value of the pathway enrichment in the mRNA data multiplied by the enrichment sign ( + 1 if positive, −1 if negative). Colour indicates whether the sign of miRNA and mRNA enrichment results for a pathway agree (blue) or disagree (red).

The numbers of pathways in each quadrant is shown on the plot. The line of best fit is shown in black with 95% confidence intervals in grey. Spearman rank (two-tailed) r = −0.04, p = 0.001. **d** Chord plot showing the cross-over between differentially expressed proteins (bright green) and mRNAs (multi-coloured), and the miRNAs (bright pink) that target them at NAAT+ve CT vs baseline in the placebo group. Track 1 (Outermost track) = feature name, track 2 = whether that feature was up (red) or downregulated (blue) at CT in NAAT+ve placebo vaccinees compared with baseline, track 3 = colour bar for that feature/feature type. The length of the feature bar corresponds to the number of links the feature has with other features. Linking lines ("chords") represent a connection between features. Links drawn between mRNA and their protein counterparts. miRNAs are connected to their targets. Links with dotted edges connect features with opposite directions of foldchange. Links without dotted edges connect features with the same direction of foldchange. Source data are provided as a Source Data file.

**Viral load data.** The NAAT test used in all participants within this study was an RT-PCR test. Cycle threshold values from the RT-PCR on nose and throat swabs (a single swab was used to swab both the nose and throat of each participant) were used as a proxy of the respiratory viral load for COVID-19 episodes. Cycle threshold values were measured for the three target SARS-CoV-2 genes − ORF1ab, S gene and N gene−and the lowest value was taken to represent the minimum cycle threshold value. For each of the COVID-19 cases, the lowest cycle threshold value from multiple swabs during the symptomatic episode was used for the analysis.

**Serum cytokine and C-reactive protein assays.** Serum was separated from whole blood samples by centrifugation at 3000 × $g$ for 10 min

and stored at −80 °C. Concentrations of the following cytokines and chemokines were measured using a custom 20-plex Luminex panel (MILLIPLEX MAP Human Cytokine/Chemokine Magnetic Bead Panel− Immunology Multiplex Assay, Merck Millipore) according to the manufacturer's instructions: interleukin (IL) 1β, IL2, IL3, IL4, IL5, IL6, IL8, IL10, IL12-p70, IL13, IL17A, IL18, soluble CD40-ligand (sCD40L), granulocyte colony-stimulating factor (G CSF), granulocyte-macrophage colony-stimulating factor (GM CSF), IFNγ, interferon-gamma induced protein 10 (IP10 or CXCL10), monocyte chemoat-tractant protein 1 (MCP1), tumour necrosis factor-alpha (TNFα) and vascular endothelial growth factor (VEGF-A). Concentrations were calculated using Luminex Xponent software. Samples were measured in duplicate, and then concentrations averaged. Samples with a

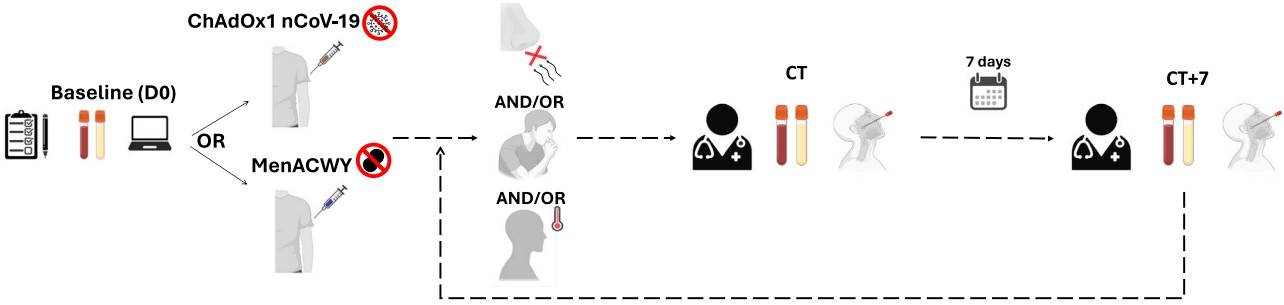

**Fig. 8 | Visit schema. Baseline visit (D0) involved assessment, enrolment and blood sampling.** Participants were then vaccinated with either ChAdOx1 nCoV-19 or placebo vaccine (MenACWY). Participants with at least one of: loss of smell or tase, continuous cough or fever attended the study site for a CT visit (COVID-19 test visit) at symptom onset for assessment, a COVID-19 NAAT test (RT-PCR test) and blood sampling. Participants presented again 7 days later (CT + 7) for repeat study procedures regardless of COVID-19 test result. Participants then returned to normal follow up until their next set of symptoms. Created with BioRender.com.

coefficient of variation value > 30% were excluded. Cytokine/chemokine concentrations below their lower limit of quantification (LLQ) were set to 50% of that limit. Cytokines detected in at least half of the samples in at least one subgroup were retained in the analysis.

To increase the sensitivity for certain cytokines, serum samples from Stage 2 were also analysed using the V-PLEX Human Pro-inflammatory Panel 1 kit (MSD), which includes IFNγ, IL1β, IL2, IL4, IL6, IL8, IL10, IL12p70, IL13, TNFα following the manufacturer's instructions. Data were generated by Methodological Mind software and analysed with MSD Discovery Workbench software. Presented data were adjusted for sample dilution.

Serum CRP was measured using a clinical-grade assay at the medical labs at Oxford University Hospital, UK.

### Statistical analysis

**Differential gene and transcript expression analysis.** Differential gene expression was conducted using the R Bioconductor packages "edgeR" (v 3.32.1) and "limma" (v 3.46.0)[79–83]. Next-gen RNA-sequencing data were normalised for RNA composition using the trimmed mean of M-value (TMM) method[82]. Data were transformed using the limma "voom" function. A linear model was fitted to the data using the limma "lmFit" function using the empirical Bayes method[81]. Differential transcript expression analysis of ONT 3rd gen RNA sequencing was performed using a similar pipeline. In the sRNA data analysis, participant was included as a random effect and red cell count abundance was included as a fixed effect where available. The cut-off for statistical significance was set at a false-discovery rate (FDR, Benjamini-Hochberg method) <0.01 for Illumina next-gen RNA sequencing and <0.05 for ONT 3rd gen RNA sequencing and sRNA sequencing. Results of differential gene expression analysis are available for each of the omics datasets are available as supplementary excel tables (Supplementary data 2, Supplementary data 3, Supplementary data 4, Supplementary data 5, Supplementary data 6, Supplementary data 7, Supplementary data 8). For the sRNA data, effect sizes were calculated by dividing an sRNA's average log$_2$ fold-change by its standard deviation.

**PCA analysis.** PCA analysis was performed using the R Bioconductor package "pcaExplorer"[84] on log-transformed data; a confidence level of 0.95 was used to capture data spread level.

**Differential serum cytokine analysis.** For analysis of cytokines, unpaired Wilcoxon tests were used to determine significant changes among different groups. FDR of <0.05 was taken as significant. Statistical analysis was performed using R (version 4.2.1).

**GSEA analysis of next generation RNA-seq data.** Gene-set enrichment analysis (GSEA) was undertaken on the entire list of filtered genes, ranked by their t-statistic from Limma, using the fgsea (v1.27.1)

R package for fast preranked GSEA[85]. Analysis was completed using the gene sets in the gene ontology biological process and the Reactome pathway databases (http://www.broadinstitute.org/gsea/msigdb/index.jsp).

**Blood transcriptional modules.** Blood transcriptional module (BTM) analysis was conducted using the "tmod" (v.0.50.13) R package on gene lists ranked by their log-ratio (LR) value; non-parametric statistical testing for module expression was undertaken using the "tmodCERNOtest" function[86].

**MiRNA set enrichment analysis.** MiRNA enrichment analyses aim to determine whether miRNA expression will likely have a suppressive or permissive effect on expression of a particular mRNA/protein or the mRNAs/proteins within a pathway. The multiple approaches used in this study are described in the supplementary methods. Unless otherwise stated, mRNA-miRNA target lists for the miRNA/mRNA integration and enrichment analyses were based on lists of experimentally validated miRNA-mRNA targets downloaded from miRTarBase v8.0, TarBase v8.0 and miRecords via miRNet[87]. miRNA-mRNA targets were filtered out if they were validated by poor-quality methods only (e.g., biotinylated arrays).

### Blood count data

Blood count data was measured using clinical complete blood count measurements obtained from sending EDTA samples to the John Radcliffe Hospital laboratory. Relative blood cell fractions were also estimated from the RNA-seq data using via the CIBERSORTx pipeline using the LM22 signature matrix[88].

### Data visualisation

Unless otherwise specified in legends, plots were rendered in R using ggplot2 (v 3.4.2). Summary plots of differential expression results are presented as volcano plots from EnhancedVolcano (v 1.8.0). Y-axes labelled -log$_{10}$(p-value) represents the log$_{10}$(raw unadjusted p-value). The contrasts used to generate the results displayed in each volcano plot can be found in Supplementary data 2.

### Study limitations

The COVID-19 cases included in this study were primarily mild, with a few of moderate severity. Thus, whether this study's results generalise to more severe illnesses is unclear. Our study's sample size is moderate but represents the largest to date from an RCT. Case-control or prospective cohort studies following patients at high risk of severe illness may be able to capture such cases in vaccinated populations but would lack the advantages of an RCT. Paired baseline samples were not available for every participant. As pairing can improve power by inherently controlling for biological noise, it is possible that some of

our differential expression results provide conservative differential expression p-value estimates. We matched all subgroups for risk factors to mitigate bias. Small differences in group sizes could potentially affect the number of differentially expressed genes in the ChAdOx1 nCoV-19 and placebo groups when compared with baseline or CT NAAT-ve samples however effect sizes and $\log_2$ foldchanges were generally larger in NAAT+ve placebo comparisons supporting the idea that perturbation in gene expression is larger during COVID-19 in those who have not received ChAdOx1 nCoV-19.

The main SARS-CoV-2 variants circulating during the study period were ancestral and Alpha, and the volunteers were naïve to prior infection at study enrolment. The ability of the ChAdOx1 nCoV-19 vaccine to protect against severe disease by other variants suggests results are generalisable to the currently circulating SARS-CoV-2 Omicron variants, but it is not clear what is the additional effect that further booster vaccine doses confer. Bulk sequencing in this study makes it challenging to separate differential expression due to intracellular changes from changes due to differences in the cellular composition of blood. However, the latter cannot account for all the differences between vaccine groups at the molecular level, as gene expression fold-changes were frequently greater than cell count fold-changes. Furthermore, CIBERSORTx inferred differential gene expression within cell populations; nevertheless, future studies employing single-cell RNA-seq or CITE-seq approaches would help improve resolution.

### Reporting summary

Further information on research design is available in the Nature Portfolio Reporting Summary linked to this article.

## Data availability

The gene expression data, such as 3rd gen long-read RNA-seq, next-gen short-read RNA-seq and small RNA-seq, generated in this study have been deposited in the Gene Expression Omnibus database under accession code GSE228842. The DGE data generated in this study are provided in the Supplementary Information. The Homo sapiens genome build GRCh38 used in this study is available in to download through HISAT2 website (https://genome-idx.s3.amazonaws.com/hisat/grch38_genome.tar.gz). Gene lists for the pathway analysis used in this study are available through Molecular Signatures Database (http://www.broadinstitute.org/gsea/msigdb/index.jsp). Micro-RNA databases used in this study are available through miRNet (https://www.mirnet.ca). Source data are provided with this paper.

## Code availability

The reproducible code for analysis is available at Zenodo (https://doi.org/10.5281/zenodo.10797098)[89] and on Github https://github.com/Chelysheva/COVID_multiomics_codes/ and https://github.com/dan-scholar/COVID_RNAseq_script/.

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

## Acknowledgements

We are grateful to the volunteers who participated in this study. We thank the Oxford Genomics Centre at the Wellcome Centre for Human Genetics (funded by Wellcome Trust grant reference 203141/Z/16/Z) for the generation and initial processing of the sequencing data. We thank the ONT team, in particular Lakmal Jayasinghe, David Stoddart, Luke McNeil, Jemma Jordan, Libby Snell, Irina Vasilescu, Mihai Manea, Steven Alcock, James Mason, and Philipp Rescheneder. We would like to thank Tomas Malinauskas for running AlphaFold. For the purpose of Open Access, the author has applied a CC BY public copyright licence to any Author Accepted Manuscript (AAM) version arising from this submission. This research was funded in part by the National Institute for Health Research (NIHR) Oxford Biomedical Research Centre (BRC) —S.G., P.K., A.J.P., D.O.C. The views expressed are those of the author(s) and not necessarily those of the NHS, the NIHR or the Department of Health. Oxford Nanopore Technologies (ONT) provided a facility and performed 3rd gen long-read RNA sequencing for no charge. The ChAdOx1 nCoV-19 randomised controlled trials were funded by the UK Research and Innovation, National Institutes for Health Research (NIHR), Coalition for Epidemic Preparedness Innovations, Bill & Melinda Gates Foundation, Lemann Foundation, Rede D'Or, Brava and Telles Foundation, NIHR Oxford Biomedical Research Centre, Thames Valley and South Midland's NIHR Clinical Research Network, and AstraZeneca. AstraZeneca reviewed the data from the study and the final manuscript before submission, but the academic authors retained editorial control. All other study funders had no role in the study design, data collection, analysis, interpretation, or report writing. All authors had full access to all the data in the study and had final responsibility for the decision to submit for publication.

## Author contributions

D.O.C., A.J.P., S.C.G. and P.K. conceived and designed this work. R.E.D., S.C., I.C. and D.O.C. performed all the bioinformatics analysis. S.B., K.S., K.E., S.F., D.P. and M.V. made substantial contributions to interpretation of data. R.E.D., S.C., I.C. and D.O.C. wrote the initial draft of the manuscript—D.M.F., T.L. and A.J.P. substantively revised it. All the authors reviewed and approved the final version of the manuscript. All authors approved the submitted version of this manuscript and have agreed both to be personally accountable for the author's own contributions and the integrity of the work.

## Competing interests

P.K. has received consultancy fees from AstraZeneca. S.C.G. is named as an inventor on the patent covering ChAdOx1 use as a vaccine vector and holds stock in Vaccitech. T.L. reports consulting fees from Vaccitech on an unrelated project, an honorarium from Seqirus, work-related investments, and is named as an inventor on a patent application for a vaccine against SARS-CoV-2. A.J.P. was a member of WHO's Strategic Advisory Group of Experts on Immunization until January, 2022 and remains chair of the UK Department of Health and Social Care's Joint Committee on Vaccination and Immunisation (JCVI) but does not participate in the JCVI COVID-19 committee; and reports providing advice to Shionogi on COVID-19, and funding from the NIHR, AstraZeneca, the Bill & Melinda Gates Foundation, Wellcome, the Medical Research Council, and the Coalition for Epidemic Preparedness Innovations. Oxford University has entered into a partnership with AstraZeneca for the development of COVID-19 vaccines.

## Additional information

Ruth E. Drury [1,2,8], Susana Camara[1,2,8], Irina Chelysheva [1,2,8], Sagida Bibi[1,2], Katherine Sanders[1,2], Salle Felle [1,2], Katherine Emary[1,2], Daniel Phillips [1,2], Merryn Voysey [1,2], Daniela M. Ferreira [1,2,3], Paul Klenerman[2,4,5], Sarah C. Gilbert [2,6,7], Teresa Lambe [1,2,7], Andrew J. Pollard [1,2,8] & Daniel O'Connor [1,2,8] ✉

[1]Oxford Vaccine Group, Department of Paediatrics, University of Oxford, Oxford, UK. [2]NIHR Oxford Biomedical Research Centre, Oxford, UK. [3]Department of Clinical Sciences, Liverpool School of Tropical Medicine, Liverpool, UK. [4]Peter Medawar Building for Pathogen Research, Nuffield Dept. of Clinical Medicine, University of Oxford, Oxford, UK. [5]Translational Gastroenterology Unit, Nuffield Department of Medicine, University of Oxford, Oxford, UK. [6]Pandemic Sciences Institute, Nuffield Department of Medicine, University of Oxford, Oxford, UK. [7]Chinese Academy of Medical Science (CAMS) Oxford Institute, University of Oxford, Oxford, UK. [8]These authors contributed equally: Ruth E. Drury, Susana Camara, Irina Chelysheva, Andrew J. Pollard, Daniel O'Connor. ✉e-mail: daniel.oconnor@paediatrics.ox.ac.uk

