## [Peer Review File · Nature Communications]

Multi-omics analysis reveals COVID-19 vaccine induced attenuation of inflammatory responses during breakthrough diseaseReviewers' Comments:

Reviewer #1:

Remarks to the Author:

In this paper, Drury et al. analyze multi-omics data from two observer-blinded randomized controlled trials of participants receiving either the ChAdOx1 vaccine or placebo, who were followed at baseline, at onset of COVID-19-like symptoms, and 7 days after. The study addresses important issues around the potential benefits of the ChAdOx1 vaccine in breakthrough infection and concludes that vaccine-mediated attenuation is taking place through abrogation of pro-inflammatory responses. We recognize the value of perspective sampling and the high potential of the datasets to address the knowledge gap surrounding vaccine benefits in breakthrough infection but overall the manuscript lacks rigor and it is particularly concerning how the authors report, present and interpret their findings and specific points are outlined below:

1- A critical example is the confusing labeling of experimental groups leading to potential misinterpretation of the results by the readers. There are 10 experimental groups in stage 1 of the study that should have been clearly labeled/presented in the figures and results section.

For example, it's not clear if the "NAAT-" labels in the figures (and "NAAT-ve" labels in the main text) refer to "NAAT- placebo", "NAAT- ChAdOx1 nCoV-19" or all NAAT- regardless of vaccine group. There seem to be 12 or 13 "CT_NAAT_neg" data points in the PCA plot in Figure 3a but what experimental group is this referring to considering that in Figure 2 there are 8 NAAT- ChAdOx1 nCoV-19 and 9 NAAT- placebo? Also the group labels are not consistent across the figures as clearly seen in most of the figures showing group contrasts. One could guess based on logic but this would lead to misinterpretation of the results and it made it extremely difficult to understand and fairly review the results of differential expression.

2- Some of the figures presented are of poor quality and unintelligible and some of the subfigure titles are missing or uninformative (e.g. Figure 5A-B and most of the labels of volcano plots). Also, there is a lack of consistency in sample sizes reported in Figures 2 with those inferred plots like PCA in Figure 3. Figure legends should include sample sizes and the lack of proper labeling of groups is not helping. Given that the omics data might not have been generated for all the stage 1 and stage 2 participants, the authors should include a supplementary table that clearly shows what data was collected and reported for each study participant (Supplementary Table 6 includes only summary data). It is also important to show in the table which datasets are matched for each participant at each stage of sample/data collection.

3- Another critical example is the presentation and interpretation of the analysis of the three transcriptomic datasets (total RNAseq, sRNAseq and long-read RNAseq). First, the authors report very little on quality control of the data and this is particularly relevant given the relatively small size of each of the experimental groups analyzed. The filtering of features done is sound but considering that linear modeling is done (it seems for all three datasets), the distribution of the transformed/normalized data should be shown for each dataset in supplementary figures. These plots will indicate the consistency of distribution among samples.

4- Enough details are not provided in the methods for the work to be reproduced. Statistical analyses of the three transcriptomic datasets in particular are not detailed enough. Important details like the number of features retained for downstream analysis are not mentioned. Code used in all data analyses reported in the paper must be provided. Importantly, the results of the analysis of each dataset should be reported separately in subsections of the Results section.

5- Lack of sRNA differential expression at CT between NAAT-ve participants (ChAdOx1 nCoV-19 ?) and NAAT+ve ChAdOx1 nCoV-19 in the stage 1 study could be due to lack of power in the stage 1 dataset. The stage 2 sRNA analysis is more powered but there are still sample size differences between placebo

and ChAdOx1 nCoV-19 (baseline vs CT) that could explain the difference in the magnitude of differential expression in the two contrasts. This possibility should be acknowledged and discussed.

6- Without functional data testing miRNA-driven attenuation of inflammatory responses, which would be challenging to do, the authors should elaborate on at least one of the potential major miRNA-mRNA pairs potentially driving the observed downstream molecular or cellular effects. The stage 2 datasets would be useful for this exercise in particular if qPCR data validates the patterns.

7- Overall there needs to be a good deal more specificity in describing the omics- analysis to support the major specific claim of paucity of post-transcriptional regulation in COVID-19 so that we may adequately and accurately refer to miRNAs and their multiple roles in their appropriate context as the one presented in this manuscript. Only by raising the level of precision in presenting the analyses and results, can the mountain of transcriptomic data that is supplied in this study be adequately described and interpreted in an immunological context.

Other comments

- At multiple instances figures and tables are referred out of order. Supplementary Tables 1 and 2 are not referenced in the manuscript
- The scale of Y axes in the volcano plots that are meant to be compared in each figure should be the same.
- In the methods it is mentioned that for the long read data transcript counts were merged by gene for differential gene expression analysis but the results report data on isoforms?
- Supplementary tables reporting the gene-level results for differentially expressed genes should be provided.

Reviewer #2:

Remarks to the Author:

In their manuscript Drury et al. describe the host response in patients, SARS-CoV-2-vaccinated or placebo-vaccinated and infected with SARS-CoV-2 virus. The manuscript addresses an extremely important question in the field: is the immune response in SARS-CoV-2-vaccinated people different from those that were not vaccinated. They performed transcriptome and proteome studies in a cohort of participants vaccinated with the ChAdOx1 vaccine and compared them with participants vaccinated with a meningococcal vaccine (MenACWY), referred to as placebo vaccine. In both groups a proportion of participants were infected with SARS-CoV-2 virus, meaning a break-through infection in SARS-CoV-2-vaccinated patients. Their results showed that vaccinated participants exhibited reduced inflammatory response with respect to transcriptome and proteome signatures. They conclude that even though vaccinated people experienced a breakthrough infection, their inflammatory response is ameliorated compared to non-vaccinated participants reducing deleterious hyperinflammation. Very importantly, they validated their findings from one cohort also in another cohort.

The manuscript is very data-rich and many analyses have been performed on different RNA species and selected proteins. It is well structured and the figures present the data well, although some aspects could be improved, see below. The discussion addresses well the findings and puts them into a general context.

The manuscript is well acceptable for publication in Nature Communication and represents an important contribution to the field. However some major revisions outlined below should be considered.

I received a revised MS word document without line numbering. I added line numbering to be able to better refer to specific statements in the text.

Line 21: please add: description of immune responses to COVID-19 in blood of. Is it responses to COVID-19 or actually responses to SARS-CoV-2 virus? I would think it is the response to the virus.

Same for line 23.

Line 32: global reduction in micro-RNA expression in vaccinated individuals? If so, please specify.

Line 87: I assume 'to receive a single dose'; please specify this.

Line 139: what where the mean number of reads per sample?

Line 140: which release of GRCH38 was used?

Line 163 ff: how many reads per sample, which mean read lengths?

Material & Methods: please describe also the GSEA analysis. When presenting the results from the GSEA analysis, it is not clear whether all genes or only a selection of genes (which thresholds) was used? Or were only the DEGs from the differential expression analysis used? For example, figure 4e and f show significant pathways but no DEGs were found for these contrasts (figure 4b and c)?

Please describe the methods for the blood count data

Line 247: please also mentioned the number of participants for the baseline

Please provide supplementary tables for all DEG comparisons, with multiple testing adjusted p-values and non-adjusted p-values. See also below my comments on volcano plots.

Figure 1 Please add the abbreviation DO to the figure, it will help the reader.

Figure 2: this is a very good overview about the cohorts, the different groups and numbers.

Nevertheless, I found it confusing that the structure was different in parts A and B. Instead of two groups red and blue in part A, one could do a group of black colored people figures for the baseline, as in part B, then text ChAdOx1 and MenACWY, then the people figures for the separate red and blue groups as in B.

Line 266: add: transcriptome in total RNA profile. Is it really total RNA or only coding RNA?

There are many instances, where more such specifications would strongly help to improve readability of the manuscript. I describe several examples below, as the one above, and I encourage the authors to thoroughly go through the entire manuscript and check where more specific descriptions would be appropriate and helpful, e.g. always mention in all figure legends: is this from total/coding RNA, miRNA, protein; is this CT or CT+7; exactly which contrasts were performed for a volcano plot

Figure 3a: It may be helpful to color vaccinated and placebo-vaccinated NAAT(-) samples with different colors; are they separating or not? When I compare the numbers of samples in this PCA with Supplementary Table S3, I can see 12 (S3 says 11) dots for the DO group, 12 dots (S3 says 13) for ChAdOx1-vaccinated NAAT(+) group, 8 dots (S3 says 9) in MenACWY-vaccinated NAAT(+) samples. I may miss some dots because they are overlapping, but I would recommend to check these numbers carefully in the PCA plot.

Figure 3: In my opinion, it does not make sense to show volcano plots with non-significant results as in c and f. It should suffice to mention that no DEGs were found in the text and provide DEG tables for genes with significant p-values and add adjusted p-values for them. Also, no need to show non-significant pathways in figure d. Figure d has an additional headline which should be removed. Fig e: although there are significant DEGs, the number is low and almost impossible to read in the volcano plot. Also here, a DEG table may suffice. I would also highly recommend to show the adjusted p-values on the Y axis instead of the non-adjusted p-values. Showing non-adjusted p-values is confusing with the red and blue dots.

The above issues apply for all other figures.

Line 290 total: please check: analysis of blood RNA-seq from total RNA, or is it actually coding RNA? Also state 'transcriptome at CT'.

Figure 4: no need to show volcano plots with non-significant DEGs, no need to show non-significant pathways in GSEA results. Figure c: adjusted p-values on y-axis would be better.

Figure 5: No need to show volcano plots with non-significant results. 5a: adjusted p-values on y-axis would be better. 5b, I can actually see the point: are the same genes regulated in vaccinated people but not significantly? However, figure c (maybe with a few more gene annotations) would do it. Figure 5c is actually a very good visualization of the effect: same genes regulated but with lower fold-change.

Figure e and f: a boxplot for the two significant genes would present the data better.

Line 319: at CT and at CT+7 ??

Line 325: GSEA from total/coding RNA?

Line 355: total or only coding RNA?

Line 368: specify cytokine protein concentration levels in blood

Figure 6: Figure b, c: adjusted p-values on y-axis would be better. Figure d: from which comparison, I guess comparison of indicated groups vs baseline, from total/coding RNA? What do the different sections of the circles show: GSEA of all genes, DEGs, or other? I guess that the figure shows only the profiler cluster analysis of xx (how many?) top pathways (by p-value or gene ratio)? Please specify.
Line 375: analysis of total or only coding RNA-seq?
Figure 7: Figure a, b: adjusted p-values on y-axis would be better. Result in figure f is very important information! Figure j: I guess the ChAdOx1 and MenACWY are only from NAAT(+) samples? Please specify.
Line 437: maybe add: of cytokine concentrations in blood
Line 441: the top three genes (by absolute fold-change)?
Line 443: Figure h: maybe add: showing clusters by cell identity and expression of ALPL and KREMEN1 in cell clusters.
Line 454: I guess that this is a different cohort? Please provide a reference.
Line 454: should read Figure 7i instead of 7I?
Line 485: figure S9 does not name specific pathways.
Line 548: add: at CT+7
Line 552: add in the upper respiratory tract
Line 554: specify that these were proteins measured
Line 584: maybe rather associated with 'faster virus clearance' than 'reduced viral load'?
Line 579: 'B cell and CD4 pathways enriched' Figure 3d nor figure S1 do show any of these pathways. Fig4 e and f show activation of these pathways in both ChAdOx1-vaccinated and placebo-vaccinated groups. Same for line 675. Also, there were no DEGs found for SARS-CoV-2 NAAT(+) samples. However, stage 2 cohort samples showed this feature (figure 6d). Maybe mention this specifically in the discussion and in the limitations. I was a bit confused.
Line 604: was this abortive or virus-producing infection in neutrophils?
Figure S1: no need to show non-significant pathways.
Figure S4: volcano and boxplot are redundant, boxplots should suffice.
Lines 1050 & 1051: cut 'and'
Line 1079: measured?
Line 1096: add: gene expression levels in neutrophils obtained from ciphersort analysis.
Line 1105: add: using the results of miRNA expression in ..
Legend figure S10: add for miRNA expression levels
Figure S11: a great figure. However, the text is difficult to read. Provide PDF instead?
Figure S13: blood count breakdown after ciphersort analysis?
Figure S14: The text is difficult to read. Provide PDF instead?

Reviewer #3:

Remarks to the Author:

Reviewer #4:

Remarks to the Author:

The paper by Drury et al (NCOMMS-23-27082-T) utilized multiomics data to investigate the effects of ChAdOx1 SARS-CoV-2 vaccine at the time of symptom onset and 7 days post symptom onset, and compared it to placebo control. After 7 days post symptom onset, the authors reported resolution of transcriptome perturbation in both vaccine arms, and enrichment of humoral immune pathways. The authors also found attenuation of gene expression in the ChAdOx1 vaccine group compared to the placebo group, and confirmed the findings in an independent cohort. Finally, the authors found that

the upregulation of inflammatory pathways at symptom onset could be due to downregulation of miRNAs that target those pathways.

The strengths of the study are the analysis of multiple data types (RNA, sRNA, absolute cell count, absolute cytokine measurement), the inclusion of placebo and symptomatic controls, and the use of an independent validation cohort.

The weaknesses are that several conclusions were not supported by the data, the statistical significance thresholds were applied inconsistently, and unclear/inaccurate figure captions that hinder interpretation of the data.

Major comments

1. There are concerning problems with how the authors determined statistical significance in the DE analyses. First, they stated that no DEGs were observed in Fig 3c (lines 271-272). However, there are a lot of genes with adjusted $P < 0.01$, which was stated as the threshold used for Illumina total RNA-seq data (line 216). Yet Fig 3b's caption stated that a threshold of $FDR < 0.05$ was used. Similarly, there are a lot of genes with adjusted $P < 0.01$ in Fig 4b-d, yet the authors claimed there were no DEGs between CT+7 and baseline (lines 306-307). This directly undermines the authors' conclusion of resolution of transcriptome perturbation.

Second, the authors used $FDR < 0.001$ for statistical significance of sRNA data in Fig 3e, f, and $FDR < 10^{-4}$ in Fig 5e, while they stated that they used $FDR < 0.05$ in the Methods section (lines 217 & 363). $FDR < 0.001$ is an unusually stringent threshold, and arguably inappropriate when the authors were emphasizing the lack of differences between the groups (lines 279-281).

2. On lines 265-266, the authors claimed that "CT samples from COVID-19 positive episodes cluster separately to CT samples from NAAT-ve individuals and baseline samples." This is not supported by Fig 3a (and also Supp Fig 3a), which shows that only the NAAT+ve placebo group appears to cluster separately from the NAAT-ve and baseline groups, while the NAAT+ve ChAdOx1 group has some overlap.

Additionally, could the authors also explain why the NAAT-ve group overlapped almost completely with the baseline group? One would expect the two groups to be different.

3. Fig 3 suggests that the NAAT-ve group contained both NAAT-ve ChAdOx1 and placebo patients. Have the authors analyzed whether the NAAT-ve placebo group has a different gene expression profile compared to NAAT-ve ChAdOx1? If they are different, that could confound the NAAT+ve placebo/ChAdOx1 vs NAAT-ve analyses. If they are not different, are the COVID-19 risk factors still comparable between the 2 groups? If they are not, those risk factors (especially age and sex) should be controlled for in the DE analyses.

4. How does the time from vaccination to CT affect the results, because vaccine's protectivity wanes with time? Within each vaccine arm, the NAAT+ve group has different time from prime/boost to s0 compared to the NAAT-ve group (Supp table 5), potentially confounding the findings.

5. In the CT/CT+7 vs baseline total RNA analyses, did the authors model the participant ID as a random effect (eg, using limma's duplicateCorrelation function)? According to the Methods, the authors only used the participant random effect in the sRNA analysis (line 214). Without taking into account correlation between samples from the same patient, the P-values could be incorrectly estimated. Ideally these analyses would only be done with patients who had baseline samples, but Fig 4a suggests that there are not enough such samples.

6. Evidence of attenuation in perturbation in stage 2 sRNA data is not convincing. The authors argued that there are more DEGs in the placebo arm compared to the ChAdOx1 arm (lines 397-399), yet the colored data points in Fig 7a, b suggest that two different significance thresholds were used. Besides the supposed different number of DEGs, how do the logFC of the two analyses compare? The authors should do an analysis similar to the one in Fig 5g.

7. The authors concluded that ChAdOx1 attenuated the molecular responses of both RNA and sRNA (of which miRNAs make up the vast majority (Supp Fig 16)). Later, they found that there was a reciprocal relationship between miRNA expression and their target mRNAs (lines 485, 486). Could the authors explore and explain this apparent contradiction?

8. The authors' statement on B cell and BCR regulation on line 547 is not supported by the data. The authors have only shown that B cell signature are enriched in the ChAdOx1 CT vs D0 comparison in Fig 6d (the direction of enrichment is unclear from the figure). Can the authors describe what Fig 6d is showing in more details? What are the modules being shown (eg, GO BP, Reactome, or something else)?
9. The authors concluded that ChAdOx1's attenuation of pro-inflammatory pathways' activity led to lower disease severity. Yet in the cohort, NAAT+ve ChAdOx1 group has more "moderate A" cases compared to the NAAT+ve placebo group. Can the authors really conclude that they have found a "mechanistic understanding" of how ChAdOx1 reduces disease severity (lines 683-684), when their cohort doesn't appear to show reduced disease severity in ChAdOx1 group?
10. The study only looked at symptomatic infections, and for good reason. Could the authors discuss whether this selection can lead to collider bias, for example with the help of a directed acyclic graph? Can the findings in the study be generalized to asymptomatic infections?

Minor comments

1. The authors show a box plot in Fig 7f comparing minimum Ct from SARS-CoV-2 RT-PCR in Figure 7 and conclude that no differences exist between groups. What is the p value? Could this conclusion be due to limited sample size?
2. How does the SARS-CoV-2 viral load at CT+7 compare to that at CT within each vaccine arm? In Fig 7f, could the authors separate stage 1 and stage 2 data in their statistical test?
3. The authors describe assaying viral load from nose and throat swabs. Given the known variability in viral load between those 2 sites, the authors should provide more details on the breakdown, and describe how the addressed anatomical sampling site bias between patients/replicates.
4. The authors interchange the terms PCR and NAAT and should choose one
5. A different abbreviation for test visit instead of CT would be better, because CT is usually associated with qPCR cycle threshold.
6. The plot title of Fig 3d (NAAT+ve (control)) doesn't match the panel label (NAAT+ve (placebo)).
7. Please describe how GSEA was done in the Methods section.
8. A different color choice for the 2 NAAT +ve groups in Fig 3a would be better, in order to avoid confusion with the up- and downregulation colors in the other Fig 3 panels.
9. A lot of the gene annotations overlap, making them difficult to read, eg, Fig 3e, 7a.
10. Please describe what "s0" in Supp Tables 5 & 6 mean in the table captions.
11. Unintended new line between lines 123 & 124.
12. Line 344 should be referring to Supp Fig 1e.
13. Fig 5a and 5b should have the same axis limits for easier comparison.
14. Fig 5a, b's titles should say NAAT+ve instead of PCRpos for consistency.
15. The horizontal dashed lines in cytokine plots should be explained in the figure captions.
16. Please describe the blood cell count data in the Methods section.
17. The link on line 443 doesn't work.
18. Which figure were the authors referring to on lines 421-423?
19. There are multiple grammatical issues with respect to the terms COVID-19 versus SARS-CoV-2. For instance, just in the abstract, the authors refer to COVID-19 disease, which is grammatically duplicative; they state that immune responses are measured in response to COVID-19 but rather it should be to SARS-CoV-2 or to the vaccine, and COVID-19 infection is incorrect- it should be SARS-CoV-2 infection
20. Typo in Figure 1 legend 'Participants were then vaccination with either (should be vaccinated)

Reviewer #5:

Remarks to the Author:

I co-reviewed this manuscript with one of the reviewers who provided the listed reports. This is part of

the Nature Communications initiative to facilitate training in peer review and to provide appropriate recognition for Early Career Researchers who co-review manuscripts.

Point-by-point response to reviewers' comments

547Reviewer #1 (Remarks to the Author):

In this paper, Drury et al. analyze multi-omics data from two observer-blinded randomized controlled trials of participants receiving either the ChAdOx1 vaccine or placebo, who were followed at baseline, at onset of COVID-19-like symptoms, and 7 days after. The study addresses important issues around the potential benefits of the ChAdOx1 vaccine in breakthrough infection and concludes that vaccine-mediated attenuation is taking place through abrogation of pro-inflammatory responses. We recognize the value of perspective sampling and the high potential of the datasets to address the knowledge gap surrounding vaccine benefits in breakthrough infection but overall the manuscript lacks rigor and it is particularly concerning how the authors report, present and interpret their findings and specific points are outlined below:

1- A critical example is the confusing labeling of experimental groups leading to potential misinterpretation of the results by the readers. There are 10 experimental groups in stage 1 of the study that should have been clearly labeled/presented in the figures and results section.

For example, it's not clear if the "NAAT-" labels in the figures (and "NAAT-ve" labels in the main text) refer to "NAAT- placebo", "NAAT- ChAdOx1 nCoV-19" or all NAAT- regardless of vaccine group. There seem to be 12 or 13 "CT_NAAT_neg" data points in the PCA plot in Figure 3a but what experimental group is this referring to considering that in Figure 2 there are 8 NAAT- ChAdOx1 nCoV-19 and 9 NAAT- placebo? Also the group labels are not consistent across the figures as clearly seen in most of the figures showing group contrasts. One could guess based on logic but this would lead to misinterpretation of the results and it made it extremely difficult to understand and fairly review the results of differential expression.

We appreciate this comment and agree on the inconsistency and complex design. NAAT-ve group refers to all the NAAT-ve participants independently of the vaccine arm allocation, this now has been clarified in the text. The figure labels throughout the manuscript have been amended accordingly to be consistent with naming of the groups, such as NAAT+ve, NAAT-ve, ChAdOx1 nCoV-19, placebo. We also amended the figure 2 to give a clearer idea of the sample numbers for each group.

2- Some of the figures presented are of poor quality and unintelligible and some of the subfigure titles are missing or uninformative (e.g. Figure 5A-B and most of the labels of volcano plots). Also, there is a lack of consistency in sample sizes reported in Figures 2 with those inferred plots like PCA in Figure 3. Figure legends should include sample sizes and the lack of proper labeling of groups is not helping. Given that the omics data might not have

been generated for all the stage 1 and stage 2 participants, the authors should include a supplementary table that clearly shows what data was collected and reported for each study participant (Supplementary Table 6 includes only summary data). It is also important to show in the table which datasets are matched for each participant at each stage of sample/data collection.

We have amended the figures and replaced the ones, which were low quality. We have included Supplementary table 7 reporting the detailed demographics of each participant, the samples and time points collected for each of the omics datasets from each participant. Figure 2 is reporting the total number of participants in each group, however, not all the omics approaches included complete set of participants. Sample size (breakdown by NAAT result, vaccine arm and time point) for each dataset is reported in Supplementary table 3.

3- Another critical example is the presentation and interpretation of the analysis of the three transcriptomic datasets (total RNAseq, sRNAseq and long-read RNAseq). First, the authors report very little on quality control of the data and this is particularly relevant given the relatively small size of each of the experimental groups analyzed. The filtering of features done is sound but considering that linear modeling is done (it seems for all three datasets), the distribution of the transformed/normalized data should be shown for each dataset in supplementary figures. These plots will indicate the consistency of distribution among samples.

The codes for preprocessing are now included to the GitHub repo. Density plots on the normalised data before and after filtering of the features for each of the sequencing datasets have been added to the supplementary (Supplementary figure 21).

4- Enough details are not provided in the methods for the work to be reproduced. Statistical analyses of the three transcriptomic datasets in particular are not detailed enough. Important details like the number of features retained for downstream analysis are not mentioned. Code used in all data analyses reported in the paper must be provided. Importantly, the results of the analysis of each dataset should be reported separately in subsections of the Results section.

Supplementary code including the versions of statistical software, packages, versions and tests have been added to the Github repositories (https://github.com/Chelysheva/COVID_multiomics_codes and https://github.com/dan-scholar/COVID_RNAseq_script), which is currently private, and access can be given to reviewers with existing Github account upon request. The repository will then become publicly available upon the publication. The number of features retained for analysis in each of the datasets are added to the methods section. We appreciate the idea of reporting the results from each of the datasets separately, however, the advantage of utilising multiomics approaches allows them to be integrated and complement each other. With a current flow of the story, we found that reporting the results by stage rather than by data type is beneficial for the reader.

5- Lack of sRNA differential expression at CT between NAAT–ve participants (ChAdOx1 nCoV-19 ?) and NAAT+ve ChAdOx1 nCoV-19 in the stage 1 study could be due to lack of power in the stage 1 dataset. The stage 2 sRNA analysis is more powered but there are still sample size differences between placebo and ChAdOx1 nCoV-19 (baseline vs CT) that could explain the difference in the magnitude of differential expression in the two contrasts. This possibility should be acknowledged and discussed.

Note vaccine type is irrelevant in the NAAT-ve group as prior receipt of the placebo or ChAdOx1 nCoV-19 vaccine would not be expected to influence gene expression in a non-COVID illness, therefore when we talk about NAAT-ve participants we mean all NAAT-ve participants combined, irrespective of vaccine group. We agree that the lack of differential expression between NAAT-ve participants and NAAT+ve ChAdOx1 nCoV-19 vaccinees in the stage 1 study could be due to lack of power as there were 2 fewer samples in the ChAdOx1 nCoV-19 group, however the magnitude of log₂ foldchanges and effect sizes were also smaller in the ChAdOx1 nCoV-19 group vs the placebo group when compared with the NAAT-ve group – these parameters are not systematically affected by sample size in the way that significance values are. This information has been added to the manuscript – Supplementary figure 19. Supplementary figure 20 has also been added showing that log₂FC and effect sizes for differentially expressed genes between NAAT+ve CT and baseline are higher in placebo vaccinees compared with ChAdOx1 nCoV-19 vaccinees in the stage II data.

We have also added the following statement to the limitations section: “Small differences in group sizes could potentially affect the number of differentially expressed genes in the ChAdOx1 nCoV-19 and placebo groups when compared with baseline or CT NAAT-ve samples however effect sizes and log₂ foldchanges were generally larger in NAAT+ve placebo comparisons supporting the idea that perturbation in gene expression is larger during COVID-19 in those who have not received ChAdOx1 nCoV-19.”

6- Without functional data testing miRNA-driven attenuation of inflammatory responses, which would be challenging to do, the authors should elaborate on at least one of the potential major miRNA-mRNA pairs potentially driving the observed downstream molecular or cellular effects. The stage 2 datasets would be useful for this exercise in particular if qPCR data validates the patterns.

Selecting/focussing on one example of a significant correlation between a miRNA and mRNA and following this up with RT-PCR work is often done in the literature; thus, we understand why the reviewer would suggest this be added to our manuscript. We have added an example of a significant negative correlation between a miRNA and an mRNA involved in inflammatory responses into our supplementary results (supplementary figure 22); however, we are not keen on this type of approach for several reasons. In the stage 2 data, 148 miRNAs and 2924 mRNAs were differentially expressed between baseline and CT. Using information on miRNA-mRNA target pairings downloaded from miRNet (accessible at

<https://www.mirnet.ca/>), there are 35,214 experimentally validated target pair interactions between the differentially expressed miRNAs and mRNAs (because many miRNAs can target one mRNA and one miRNA can target many mRNAs). We feel our systems-level approach is more robust than the reviewer's suggestion of picking a significant miRNA-mRNA target pair correlation out of a possible 35,214 and doing RT-PCR to back up that correlation. The methodology of RNA sequencing is widely accepted in the literature as providing accurate information (as opposed to older methods like microarray, which did require RT-PCR validation). In addition, we have previously shown that the widely used Illumina-based sRNA-sequencing and statistical approach (e.g. limma) used in our analyses generate reliable data through previous RT-PCR confirmatory work in other studies (see Drury, R. E. Characterisation of Small RNAs in Response to Human Vaccination and Infection. University of Oxford, 2023.) Given this, we do not feel that additional RT-PCR data backing up correlations in the sequencing data adds sufficient information to this study, particularly given that our integration analyses are based on miRNA-mRNA target pairs that have already been experimentally validated. It is not a way to prove our systems-wide observation of the relationship between global trends in miRNA and mRNA expression. To do this, you would have to measure most miRNAs and mRNAs by RT-PCR, which is not viable.

7- Overall there needs to be a good deal more specificity in describing the omics- analysis to support the major specific claim of paucity of post-transcriptional regulation in COVID-19 so that we may adequately and accurately refer to miRNAs and their multiple roles in their appropriate context as the one presented in this manuscript. Only by raising the level of precision in presenting the analyses and results, can the mountain of transcriptomic data that is supplied in this study be adequately described and interpreted in an immunological context.

We appreciate the reviewer's comments regarding our data suggesting a possible link between a reduction of miRNA-mediated suppression during acute COVID-19 and widespread upregulation of pro-inflammatory genes at the mRNA and protein level. Our paper presents patterns that emerge from integrating the different omics results. We present a logical potential interpretation of those patterns, opening new hypotheses to be tested in future studies. The fact that we see a global downregulation in miRNA expression and a reciprocal upregulation in the mRNAs (and pathways), backed up by statistical inference, aligns with a potential pathological role of miRNA suppression in COVID-19. We feel we have been suitably conservative in our communication of the results. We do not assert/claim that our data *proves* global changes in miRNA expression drive pathological inflammation in COVID-19. We simply say that the data is in line with such a phenomenon.

Our system's approach (possible via gene set enrichment analyses) simultaneously evaluates the expression of all mRNA and pathways with respect to the expression of all miRNAs. This is the best way to interrogate whether changes in miRNA expression in SARS-CoV-2 infection could be linked to the widespread upregulation of pro-inflammatory genes at the mRNA and

protein level. As noted in our answer above, depicting a handful of significant miRNA-mRNA correlations cannot do this, whereas the systems-level analysis can.

It stands to reason that the significant reduction in miRNA expression at the \log_2 foldchange level (see figure 8a) reduces global miRNA mediated suppression, and this would be expected to impact on (upregulate/enable the upregulation of/ remove buffering of) mRNA and protein expression. This is in keeping with how miRNA and mRNA relate to each other.

Our data supports this in COVID-19:

- Figure 7c shows a reciprocal relationship between changes in mRNA expression and the expression of miRNAs that target those mRNAs at CT (figure 7b); this was also true at the pathway level (figure 7c).
- Figure 8 b shows that mRNAs that are targeted by downregulated miRNAs are generally upregulated.
- Figure 8c shows that pathways predicted to be released from repression based on changes in miRNA expression at CT are indeed upregulated at the mRNA level at CT, and vice versa, and this is present at the global level from a statistical perspective. Figure 15 shows a similar reciprocal relationship at CT+7.
- Figure 8d gives examples of reciprocal relationships between changes in the expression of miRNAs and key pro-inflammatory mRNA and proteins which those miRNAs are known to target – this is similar to what the reviewer wants to see in their earlier comment.

Given that many genes dysregulated in COVID-19 are linked to the inflammatory response, it is logical to suggest that the observed changes in the expression of miRNAs that target those genes may contribute to the inflammatory response. We provide several figures which support such a hypothesis:

- Supplementary Figure 9 shows increased targeting of immune pathways in NAAT+ve participants compared with participants with COVID-19, potentially indicating a paucity of post-transcriptional regulation in COVID-19 infection compared with other infections or illnesses.
- Supplementary figure 10 provides examples of immune pathways which are predicted to be de-repressed during COVID-19 compared with health owing to a significant reduction in the expression of miRNAs that target those pathways.
- Supplementary Figure 14 provides examples of immune pathways predicted to be released from miRNA repression during COVID-19 at symptom onset compared with 7 days later (as infection resolves - NB most infections were mild/moderate in our study). The results reflect the reduction in the expression of miRNAs that target immune pathways at CT, with those miRNAs beginning to return to normal

levels at CT+7. The GSEA style approach can be challenging to understand at first glance for people who are not used to interpreting such analyses – for this reason, we provide some examples in supplementary fig 14c showing how a systematic reinstatement in the expression of miRNAs that target pro-inflammatory pathways start to occur at CT+7

We recognise that the use of miRNA-target enrichment analyses can be difficult to intuitively interpret to those not used to using the methodology; therefore, we provide an extensive explanation of how such analyses are carried out and interpreted in our supplementary methods section (see supplementary methods: miRNA enrichment analyses).

Suggesting that a global reduction in miRNA expression may contribute to the overt pro-inflammatory state in COVID-19 is backed up by *in-vitro* and *in-vivo* work in the literature. For example, Igoillo-Esteve et al, 2022 (PMID: 36555120) review points out “inflammation is primarily regulated by miRNAs through their altered expression in certain immune cells. As a part of the inflammatory response, the biogenesis of miRNAs is often regulated at different stages, such as the synthesis, processing, and stabilisation of pre- or mature miRNAs”. Aguado et al (<https://doi.org/10.1016/j.chom.2015.11.003>) depletion in primary cell culture demonstrated that miRNA depletion specifically enhances cytokine expression through a model of miRNA depletion in primary cell lines. Steiner et al. (<https://doi.org/10.1016/j.immuni.2011.07.009>) showed that a global miRNA deficiency is associated with increased IFN- γ expression. A review by De Cauwer et al. (<https://doi.org/10.3389/fimmu.2018.01647>) summarises the link between diminished DICER expression (key to miRNA biogenesis) and excessive inflammation in rheumatoid arthritis. Wu et al. (doi: 10.7150/thno.41894) show global miRNA deficiency through DICER knockdown promotes inflammation at the cellular level. In addition, Garnier et al. (PMID: 35696619) linked greater global miRNA downregulation to greater COVID-19 severity based on nasopharyngeal swab samples. We have now added references to these studies in our discussion.

As the reviewer suggests, further work can be done to definitively prove a causal link between global changes in miRNA expression and inflammation, but that work would be of such a complex nature that it would warrant a separate paper. We are open to pursuing this in future. We have added the following to our limitations section: “Functional work is required to prove causation between miRNA expression and inflammation in COVID-19, nevertheless such a possibility is supported by multiple studies showing that global miRNA reduction is associated with increased cytokine production and excessive inflammation. Our results open the door to further research on whether reinstating global or specific miRNA levels could ameliorate COVID-19 severity.”

Other comments

- At multiple instances figures and tables are referred out of order. Supplementary Tables 1 and 2 are not referenced in the manuscript

Figures were reordered to be in the correct order of referencing where appropriate, Supplementary tables are now also referenced in the manuscript.

- The scale of Y axes in the volcano plots that are meant to be compared in each figure should be the same.

We appreciate this comment, however since we utilised multiple approaches, figures were scaled according to the level of perturbation observed and single scale would reduce the readability of the plots with less significant levels, therefore the default settings for the representation are used in each case.

- In the methods it is mentioned that for the long read data transcript counts were merged by gene for differential gene expression analysis but the results report data on isoforms?

We agree with this mistake, the analysis of ONT sequencing by gene was performed as well, however, we report only findings from the transcript/isoform level in this manuscript. The methods section has been amended accordingly.

- Supplementary tables reporting the gene-level results for differentially expressed genes should be provided.

These tables are now added as supplementary (Supplementary tables 8 – 12).

Reviewer #2 (Remarks to the Author):

In their manuscript Drury et al. describe the host response in patients, SARS-CoV-2-vaccinated or placebo-vaccinated and infected with SARS-CoV-2 virus. The manuscript addresses an extremely important question in the field: is the immune response in SARS-CoV-2-vaccinated people different from those that were not vaccinated. They performed transcriptome and proteome studies in a cohort of participants vaccinated with the ChAdOx1 vaccine and compared them with participants vaccinated with a meningococcal vaccine (MenACWY), referred to as placebo vaccine. In both groups a proportion of participants were infected with SARS-CoV-2 virus, meaning a break-through infection in SARS-CoV-2-vaccinated patients. Their results showed that vaccinated participants exhibited reduced inflammatory response with respect to transcriptome and proteome signatures. They conclude that even though vaccinated people experienced a breakthrough infection, their inflammatory response is ameliorated compared to non-vaccinated participants reducing deleterious hyperinflammation. Very importantly, they validated their findings from one cohort also in another cohort.

The manuscript is very data-rich and many analyses have been performed on different RNA species and selected proteins. It is well structured and the figures present the data well, although some aspects could be improved, see below. The discussion addresses well the

findings and puts them into a general context.

The manuscript is well acceptable for publication in Nature Communication and represents an important contribution to the field. However some major revisions outlined below should be considered.

I received a revised MS word document without line numbering. I added line numbering to be able to better refer to specific statements in the text.

Line 21: please add: description of immune responses to COVID-19 in blood of. Is it responses to COVID-19 or actually responses to SARS-CoV-2 virus? I would think it is the response to the virus. **Done**

Same for line 23. **As COVID-19 represents the disease caused by SARS-CoV-2 (which was the case in all our participants as all had symptomatic infections) we have used the terms somewhat interchangeably**

Line 32: global reduction in micro-RNA expression in vaccinated individuals? If so, please specify. **All participants – both ChAdOx1 nCoV-19 and placebo-vaccinated adults had a global reduction in miRNA expression in COVID-19. There is no obvious biological reason that the receiving a meningitis vaccine up to 6 months before a SARS-CoV-2 infection (the placebo group) causes a global downregulation of miRNA during SARS-CoV-2 infection so we generalise this result to people with acute COVID-19 regardless of vaccine status**

Line 87: I assume 'to receive a single dose'; please specify this. **We have added the following sentence to clarify this in the text. "The original protocol involved a single dose schedule, however an amendment was made to offer a booster dose (ChAdOx1 nCoV-19 / placebo) was implemented from 3rd August 2020. Participants second vaccination was the same as their initial vaccine. More information is contained in the original COV1/COV2 study paper."**

Line 139: what where the mean number of reads per sample? **Supplementary table 2 has been added with this information in**

Line 140: which release of GRCh38 was used? **We have added this information. See sentence: "Gene features were counted using HTSeq (version 0.11.1), using human gene annotation general transfer format version GRCh38.92"**

Line 163: how many reads per sample, which mean read lengths? **This information is contained in Supplementary table 2. We have signposted readers to this at the end of the first results section.**

Material & Methods: please describe also the GSEA analysis. When presenting the results from the GSEA analysis, it is not clear whether all genes or only a selection of genes (which thresholds) was used? Or were only the DEGs from the differential expression analysis used? For example, figure 4e and f show significant pathways but no DEGs were found for these contrasts (figure 4b and c)? **To summarise, Gene-set enrichment analysis (GSEA) was undertaken on the entire list of filtered genes, ranked by their t-statistic from Limma, using the fgsea (v1.27.1) R package for fast preranked GSEA 33. We have added a methods section on GSEA methodology.**

Please describe the methods for the blood count data, Details added

Line 247: please also mentioned the number of participants for the baseline Details added
Please provide supplementary tables for all DEG comparisons, with multiple testing adjusted p-values and non-adjusted p-values. See also below my comments on volcano plots. We have now added supplementary results files which are excel files that contain all the results tables which we created in the differential gene expression analyses.

Figure 1 Please add the abbreviation DO to the figure, it will help the reader. The figure has been updated

Figure 2: this is a very good overview about the cohorts, the different groups and numbers. Nevertheless, I found it confusing that the structure was different in parts A and B. Instead of two groups red and blue in part A, one could do a group of black colored people figures for the baseline, as in part B, then text ChAdOx1 and MenACWY, then the people figures for the separate red and blue groups as in B. The figure has been updated accordingly

Line 266: add: transcriptome in total RNA profile. Is it really total RNA or only coding RNA? Added as suggested. Thank you for pointing out this confusion. We used a ribodepletion method for the next-gen RNA seq in which both coding and non coding RNA was analysed (i.e. total RNA) – we have amended the paper to call this next-gen RNA-seq to differentiate this from the small-RNA seq and the long read RNA seq datasets.

There are many instances, where more such specifications would strongly help to improve readability of the manuscript. I describe several examples below, as the one above, and I encourage the authors to thoroughly go through the entire manuscript and check where more specific descriptions would be appropriate and helpful, e.g. always mention in all figure legends: is this from total/coding RNA, miRNA, protein; is this CT or CT+7; exactly which contrasts were performed for a volcano plot Legends amended as suggested and a supplementary 7 added which also summarises which data set and contrast were used to generate each volcano plot figure

Figure 3a: It may be helpful to color vaccinated and placebo-vaccinated NAAT(-) samples with different colors; are they separating or not? We feel that this would lead to too many colours in the plot, making it difficult to read/interpret. Also the vaccine received by people who did not have a COVID-19 infection (NAAT -ve participants) is irrelevant – we do not expect vaccine specific differences amongst NAAT-ve participants since their allocation to vaccine group was blinded and random and neither the ChAdOx1 nCoV-19 nor the placebo vaccine would have offered protection against the illness they developed, and they had been

When I compare the numbers of samples in this PCA with Supplementary Table S3, I can see 12 (S3 says 11) dots for the DO group - Table S3 says 11 samples for D0 group (black dots on PCA), there are 10 dots on the PCA - S3 table was incorrect and has been updated
12 dots (S3 says 13) for ChAdOx1-vaccinated NAAT(+) group Table S3 says 7 samples in the for ChAdOx1-vaccinated NAAT(+) (red dots on PCA) and there are 7 dots in the PCA
8 dots (S3 says 9) in MenACWY-vaccinated NAAT(+) samples there are 9 dots (blue dots) in the PCA for the MenACWY-vaccinated NAAT(+) samples

I may miss some dots because they are overlapping, but I would recommend these numbers carefully in the PCA plot – there are the correct number of dots on the PCAs - see comment above

Figure 3: In my opinion, it does not make sense to show volcano plots with non-significant results as in c and f. It should suffice to mention that no DEGs were found in the text and provide DEG tables for genes with significant p-values and add adjusted p-values for them. Also, no need to show non-significant pathways in figure d. We have removed the plots as suggested by the reviewer

Figure d has an additional headline which should be removed. Additional headline was removed

Fig e: although there are significant DEGs, the number is low and almost impossible to read in the volcano plot. Also here, a DEG table may suffice. We have removed the plots as suggested by the reviewer

I would also highly recommend the adjusted p-values on the Y axis instead of the non-adjusted p-values. Showing non-adjusted p-values is confusing with the red and blue dots. Many thanks for your comment – whilst we recognise some authors present volcano plots like this, we have preference for showing raw p-values on the y-axis of volcano plots and labelling/highlighting FDR significant results in colour as this is in line with Gordon Smyth's (limma package developer) advice as visual information is lost when plotting FDR values on y-axis due to different raw p-values sometimes being assigned to the same FDR (see Prof Smyth's advice at this link: [Volcano plot labeling troubles \(bioconductor.org\)](https://www.bioconductor.org/packages/2.14/bioc/vignettes/limma/doc/inst/doc/volcano.html)). In addition, given the distribution of FDR p-values differs to the distribution of the raw-p-values, using FDR on the y-axis can lead to odd squashed looking volcano plots that is hard to compare with another volcano plot. As you are aware, $p=0.000125$ may be FDR significant in one comparison but not FDR significant in another comparison as the Benjamini Hochburg method decides an FDR based on the distribution of raw p-values – which may be different for different comparisons. The colours of the dots make it easier to communicate in the legend what the results indicate e.g. red upregulated in group "A" at $FDR < 0.05$, blue in group "B" at $FDR < 0.05$, grey not differentially expressed at $FDR < 0.05$.

Line 290 total: please check: analysis of blood RNA-seq from total RNA, or is it actually coding RNA? We recognise total RNA sounds ambiguous when we have three different RNA-seq strategies as it could imply the results of all three RNA-seq strategies together. As both coding and non-coding RNA was included in the analysis technically "total" RNA was measured - but it was done using next-gen library prep - we have therefore deprecated the term total RNA for the term next-gen RNA-seq

state 'transcriptome at CT'. Done

Figure 4: no need to show volcano plots with non-significant DEGs, no need to show non-significant pathways in GSEA results. Volcano plots with non-significant DEGs have been moved to supplementary; non-significant pathways from GSEA have been removed from the plots.

Figure c: adjusted p-values on y-axis would be better. See our reply higher up about using raw p-values for y-axes.

Figure 5: No need to show volcano plots with non-significant results. Volcano plots with non-significant DEGs have been moved to supplementary.

5a: adjusted p-values on y-axis would be better. See our reply higher up about using raw p-values for y-axes.

5b, I can actually see the point: are the same genes regulated in vaccinated people but not significantly? However, figure c (maybe with a few more gene annotations) would do it. Figure 5c is actually visualization of the effect: same genes regulated but with lower fold-change. We have removed figure b) as there were no FDR significant DE genes, and as you say, figure c) is sufficient.

Figure e and f: a boxplot for the two significant genes would present the data better. Figure e volcano plot replaced with box plot instead (figure 5d) as reviewer suggested. Figure f removed as no significant results

Line 319: at CT and at CT+7 ?? D0 vs CT+7 – legend updated to make this clearer

Line 325: GSEA from total/coding RNA? Next-gen RNA seq (coding and non-coding) legend updated with term next-gen RNA seq.

Line 355: total or only coding RNA? Next-gen RNA seq (coding and non-coding) legend updated with term next-gen RNA seq.

Line 368: specify cytokine protein concentration levels in blood. legend states serum cytokine concentrations

Figure 6: Figure b, c: adjusted p-values on y-axis would be better. See our reply higher up about using raw p-values for y-axes.

Figure d: from which comparison, I guess comparison of indicated groups vs baseline, from total/coding RNA? Yes – from results of next-gen RNA-seq

What do the different sections of the circles show: GSEA of all genes, DEGs, or other? I guess that the figure shows only the profiler cluster analysis of xx (how many?) top pathways (by p-value or gene ratio?)? Please specify. An explanation has been added to the legend.

Line 375: analysis of total or only coding RNA? - Results of next-gen RNA-seq – legend updated.

Figure 7: Figure a, b: adjusted p-values on y-axis would be better. Please, see our reply higher up about using raw p-values for y-axes.

Result in figure f is very important information! We agree

Figure j: I guess the ChAdOx1 and MenACWY are only from +ve samples? Please specify. Yes they are NAAT +ve samples. Figure amended to reflect this.

Line 437: maybe add: of cytokine concentrations in blood. These were serum (rather than whole blood) cytokine levels.

Line 441: the top three genes (by absolute fold-change)? By significance – the legend has been updated to indicate this

Line 443: Figure h: maybe add: showing clusters by cell identity and expression of ALPL and KREMEN1 in cell clusters. Updated as per your suggestion

Line 454: I guess that this is a different cohort? Please provide a reference. This is a complete cohort from the vaccine trial to that date ([https://doi.org/10.1016/S0140-6736\(20\)31604-4](https://doi.org/10.1016/S0140-6736(20)31604-4)), which also included the participants from whom the data for this study was collected.

Reference of the original publication has been added accordingly.

Line 454: should read Figure 7i instead of 7I? yes – corrected thanks

Line 485: figure S9 does not name specific pathways. Results file added with pathway names and associated enrichment statistics – see Stage 1 MIEAA2 GO_BP results tables.xlsx

Line 548: add: at CT+7 Updated as per reviewer suggestion

Line 552: add in the upper respiratory tract - Updated as per reviewer suggestion

Line 554: specify that these were proteins measured Updated as per reviewer suggestion

Line 584: maybe rather associated with 'faster virus clearance' than 'reduced viral load'? Updated as per reviewer suggestion

Line 579: 'B cell and CD4 pathways enriched' Figure 3d nor figure S1 do show any of these pathways. Fig4 e and f show activation of these pathways in both ChAdOx1-vaccinated and placebo-vaccinated groups. The statement was based on the GSEA results in the stage 2 (larger, better powered) cohort as shown in Fig4 e and f

Same for line 675. This line number is in the conclusion and does not relate to enrichment results – is the reviewer referring to a different line?

Also, there were no DEGs found for SARS-CoV-2 NAAT(+) samples. However, stage 2 cohort samples showed this feature (figure 6d). Maybe mention this specifically in the discussion and in the limitations. I was a bit confused. We are assuming that the reviewer is asking how a pathway can be enriched yet genes within that pathway not reach the significance threshold for differential expression. The GSEA gene module analysis uses the entire gene list output of a differential expression analysis with that gene list ranked by the t-test statistic (regardless of whether a gene's p-value was less than 0.05). This type of analysis is very powerful because it is based on the overall pattern of where a pathway's genes fall within that list. For example, if the majority of a pathway's genes lie at the top of the list – i.e. are upregulated (even if some/all of those genes are not called differentially expressed) then that pathway is positively enriched. There can be some discrepancies between the

enrichment results for the stage I and II data – this may be partially explained by the larger sample size in the stage II data which means that the ranking of a gene within a list is less subject to random fluctuation arising from the impact of a single/few random extreme sample values which can have a large influence when dealing with small sample sizes. Sample size limitations discussed in discussion.

Line 604: was this abortive or virus-producing infection in neutrophils? It's possible but no studies support this to date, therefore not proffered as a possibility

Figure S1: no need to show non-significant pathways. We have removed these as the reviewer suggested

Figure S4: volcano and boxplot are redundant, boxplots should suffice. We have amended as the reviewer suggested

Lines 1050 & 1051: cut 'and' typo corrected as per reviewer suggestion

Line 1079: measured? typo corrected as per reviewer suggestion

Line 1096: add: gene expression levels in neutrophils obtained from ciphersort analysis. Text updated as per reviewer suggestion

Line 1105: add: using the results of miRNA expression in .. Text updated as per reviewer suggestion

Legend figure S10: add for miRNA expression levels Text updated as per reviewer suggestion

Figure S11: a great figure. However, the text is difficult to read. Provide PDF instead? Figure resolution improved

Figure S13: blood count breakdown after ciphersort analysis? measured by clinical complete bloods counts – legend updated to include this information

Figure S14: The text is difficult to read. Provide PDF instead? When final figures are uploaded as separate files as per formatting guidelines for final upload the resolution will come out better.

I co-reviewed this manuscript with one of the reviewers who provided the listed reports. This is part of the Nature Communications initiative to facilitate training in peer review and to provide appropriate

Reviewer #4 (Remarks to the Author):

The paper by Drury et al (NCOMMS-23-27082-T) utilized multiomics data to investigate the effects of ChAdOx1 SARS-CoV-2 vaccine at the time of symptom onset and 7 days post symptom onset, and compared it to placebo control. After 7 days post symptom onset, the authors reported resolution of transcriptome perturbation in both vaccine arms, and enrichment of humoral immune pathways. The authors also found attenuation of gene expression in the ChAdOx1 vaccine group compared to the placebo group, and confirmed the findings in an independent cohort. Finally, the authors found that the upregulation of inflammatory pathways at symptom onset could be due to downregulation of miRNAs that target those pathways.

The strengths of the study are the analysis of multiple data types (RNA, sRNA, absolute cell count, absolute cytokine measurement), the inclusion of placebo and symptomatic controls, and the use of an independent validation cohort.

The weaknesses are that several conclusions were not supported by the data, the statistical significance thresholds were applied inconsistently, and unclear/inaccurate figure captions that hinder interpretation of the data.

Major comments

Many of the major comments refer to the statistical significance which comes from the reviewer misreading of the volcano plot y-axis. Significance thresholds were consistently applied as per the methods section: i.e. “The cut-off for statistical significance was set at a false-discovery rate (FDR, Benjamini-Hochberg method) <0.01 for next-gen RNA sequencing and <0.05 for ONT 3rd gen RNA sequencing and sRNA sequencing.” “For analysis of cytokines....FDR (Benjamini-Hochberg method) of <0.05 was taken as significant.” The y-axes of the volcano plots are $-\log_{10}(\text{p-value})$ where p-value means the raw p-value not $-\log_{10}$ - (adjusted p-value) which I think is what the reviewer thought. We have indicated which genes were FDR significant on the graphs using colour as per the figure legends. We have added information to our methods section to highlight to readers what our y-axis represents in volcano plots.

Whilst we recognise some authors present volcano plots with y axis showing $-\log_{10}(\text{adjusted p-value})$, we prefer to show raw p-values on the y-axis of volcano plots and labelling/highlighting FDR significant results in colour as this is in line with Gordon Smyth’s (limma package developer) advice. This is because visual information is lost when plotting FDR values on y-axis due to different raw p-values sometimes being assigned to the same FDR (see Prof Smyth’s advice at this link: <https://support.bioconductor.org/p/62384/>). In addition, given the distribution of FDR p-values differs to the distribution of the raw-p-values, using FDR on the y-axis can lead to an odd squashed looking volcano plots that is hard to compare with another volcano plot. For example, $p=0.000125$ may be FDR significant in one comparison but not FDR significant in another comparison as the Benjamini Hochburg method decides an FDR based on the distribution of raw p-values – which may differ between comparisons.

comments in th

1. There are concerning problems with how the authors determined statistical significance in the DE analyses. First, they stated that no DEGs were observed in Fig 3c (lines 271-272). However, there are a lot of genes with adjusted $P<0.01$, which was stated as the threshold used for Illumina total RNA-seq data (line 216). Yet Fig 3b’s caption stated that a threshold of $\text{FDR}<0.05$ was used. Similarly, there are a lot of genes with adjusted $P<0.01$ in Fig 4b-d, yet the authors claimed there were no DEGs between CT+7 and baseline (lines 306-307). This directly undermines the authors’ conclusion of resolution of transcriptome perturbation.

Second, the authors used $FDR < 0.001$ for statistical significance of sRNA data in Fig 3e, f, and $FDR < 10^{-4}$ in Fig 5e, while they stated that they used $FDR < 0.05$ in the Methods section (lines 217 & 363). $FDR < 0.001$ is an unusually stringent threshold, and arguably inappropriate when the authors were emphasizing the lack of differences between the groups (lines 279-281). The reviewer has misread the y-axis title on the volcano plots which we labelled $-\log_{10}(\text{p-value})$ not $-\log_{10}(\text{FDR})$. Therefore the reviewer's interpretation of significance thresholds are incorrect. We have used consistent significance thresholds as per our methods section. Please see explanatory comment above.

2. On lines 265-266, the authors claimed that "CT samples from COVID-19 positive episodes cluster separately to CT samples from NAAT-ve individuals and baseline samples." This is not supported by Fig 3a (and also Supp Fig 3a), which shows that only the NAAT+ve placebo group appears to cluster separately from the NAAT-ve and baseline groups, while the NAAT+ve ChAdOx1 group has some overlap. Additionally, could the authors also explain why the NAAT-ve group overlapped almost completely with the baseline group? One would expect the two groups to be different.

The centre (i.e. centroid) of the NAAT+ve ChAdOx1 nCoV-19 cluster is distinctly separate to the baseline and NAAT-ve samples, however we appreciate that 1 baseline sample (out of 9) overlaps with the red samples. We have shown the 95% confidence intervals of the data on the PCA, rather than the 95% confidence interval of the cluster's centroid which we appreciate makes it hard for the reader to see how the average location (i.e. cluster centre) of the NAAT+ve samples is statistically different to that of the NAAT-ve and baseline samples. We have updated our text as per reviewer suggestion that NAAT+ve placebo group appears to cluster separately from the NAAT-ve and baseline groups.

The PCA shows that the genes which show the greatest variation across all samples were those that are expressed differently between the NAAT+ve versus baseline and NAAT-ve groups. This reflects the fact that COVID-19 infection induces a COVID-19 specific signature which is not induced by non-COVID-19 illness.

Because the NAAT-ve group represents a heterogeneous set of diagnoses each of which may induce disparate gene expression patterns, we would not necessarily expect large, consistent differences in gene expression between NAAT-ve and baseline samples. The lack of common/consistent gene expression differences between NAAT-ve and baseline samples means that the most variable genes (which the PCA algorithm identifies and upweights) are not systematically differently expressed between NAAT-ve and baseline samples. Instead the most variable genes amongst samples were those consistently induced/repressed during COVID-19 infection.

3. Fig 3 suggests that the NAAT-ve group contained both NAAT-ve ChAdOx1 and placebo patients. Have the authors analyzed whether the NAAT-ve placebo group has a different gene expression profile compared to NAAT-ve ChAdOx1? If they are different, that could

confound the NAAT+ve placebo/ChAdOx1 vs NAAT-ve analyses. If they are not different, are the COVID-19 risk factors still comparable between the 2 groups? If they are not, those risk factors (especially age and sex) should be controlled for in the DE analyses. As we have selected samples from an RCT the randomisation and blinding process means there ought not to be any systematic differences between the ChAdOx1 nCoV-19 and placebo groups in the NAAT-ve arms. This is the key advantage of basing this sub-study within an RCT. As NAAT-ve participants did not have COVID-19 or meningococcal disease the vaccine history of NAAT-ve participants is essentially irrelevant.

We appreciate that random differences between groups can still arise but random differences between ChAdOx1 nCoV-19 and placebo NAAT-ve groups does not translate to systematic differences between ChAdOx1 nCoV-19 and placebo NAAT+ve groups as each subgroup was selected independently. Furthermore we matched the participants in each subgroup for risk factors so we can assure the COVID-19 risk factors are comparable between all subgroups (see supplementary tables 5 and 6).

4. How does the time from vaccination to CT affect the results, because vaccine's protectivity wanes with time? There is no connection between the time since vaccination to CT and the molecular response. Principal component analysis figures highlighting the time since last vaccination (only >14 days were considered for the vaccine to take an effect) has been added to supplementary figures 1i for stage 1 and 4d for stage 2 next-gen RNA-seq.

Within each vaccine arm, the NAAT+ve group has different time from prime/boost to s0 compared to the NAAT-ve group (Supp table 5), potentially confounding the findings.

Given the NAAT-ve groups did not have COVID nor meningococcal disease, and given they were blinded to their group allocation, their vaccine history is essentially irrelevant – it does not matter when they received their study vaccine as the vaccine would not have been protective against the non COVID illness they developed. The waning of ChAdOx1 nCoV-19 / men-ACWY immunity in the NAAT-ve group is irrelevant to the gene expression profile they mounted against a non COVID illness and thus there is no mechanism by which the difference in time between vaccine and CT could confound results.

5. In the CT/CT+7 vs baseline total RNA analyses, did the authors model the participant ID as a random effect (eg, using limma's duplicateCorrelation function)? According to the Methods, the authors only used the participant random effect in the sRNA analysis (line 214). Without taking into account correlation between samples from the same patient, the P-values could be incorrectly estimated. Ideally these analyses would only be done with

patients who had baseline samples, but Fig 4a suggests that there are not enough such samples.

We agree that having 100% paired baseline and CT/CT+7 samples would have been ideal but it was not possible as baseline samples stopped being collected in the RCT. Not being able to include pairing information in some of our analyses means that interindividual variability cannot easily be included, this is a limitation we have added to our discussion section.

6. Evidence of attenuation in perturbation in stage 2 sRNA data is not convincing. The authors argued that there are more DEGs in the placebo arm compared to the ChAdOx1 arm (lines 397-399), yet the colored data points in Fig 7a, b suggest that two different significance thresholds were used – this statement is incorrect – the same FDR significance threshold was used. The reviewer has misread the y-axis title on the volcano plots which we labelled $-\log_{10}(\text{p-value})$ not $-\log_{10}(\text{FDR})$. Therefore the reviewer's comment that different significance thresholds were used is not correct.

Besides the supposed different number of DEGs, how do the logFC of the two analyses compare? The authors should do an analysis similar to the one in Fig 5g.

We agree figures like Fig 5g do illustrate differences in log2 FC nicely – we have therefore added a similar figure for the sRNA data in supplementary figure 17 which shows that the log2 FC and effect sizes were generally larger (i.e. there was greater perturbation in gene expression) in the placebo CT vs baseline analysis compared with the ChAdOx1 nCoV-19 CT vs baseline analysis.

7. The authors concluded that ChAdOx1 attenuated the molecular responses of both RNA and sRNA (of which miRNAs make up the vast majority (Supp Fig 16)). Later, they found that there was a reciprocal relationship between miRNA expression and their target mRNAs (lines 485, 486). Could the authors explore and explain this apparent contradiction?

The observation that prior ChAdOx1 vaccination attenuated changes (i.e. led to fewer/smaller changes) in both sRNA and mRNA expression does not contradict the observation that there was a reciprocal relationship between miRNA expression and their target mRNA; indeed if anything, the two observations support each other. The more downregulated a miRNA is, the more upregulated its mRNA is – and this occurred even more so in the placebo group (where larger changes in both sRNA and mRNA was seen). Had ChAdOx1 nCoV-19 caused larger changes in sRNA expression but smaller changes in mRNA expression at CT then *that* would have been a contradiction.

8. The authors' statement on B cell and BCR regulation on line 547 is not supported by the data. The authors have only shown that B cell signature are enriched in the ChAdOx1 CT vs D0 comparison in Fig 6d (the direction of enrichment is unclear from the figure). Can the

authors describe what Fig 6d is showing in more details? What are the modules being shown (eg, GO BP, Reactome, or something else)?

We assume the reviewer is referring to the following statement: “Interestingly, we described increased expression of genes involved in B cells and BCR regulation in COVID-19 cases compared with COVID-19-like illness, a feature recently described in a prospective study comparing COVID-19 and influenza patients”. We agree this is confusing so have removed the statement.

We have added an explanation of what figure 6d is showing, along with an explanation of blood transcriptional modules. Added text: “Blood transcriptional modules (BTM) enriched during COVID-19 compared with baseline. Enriched BTMs (FDR < 0.001) are displayed. Segments of the pie charts represent the proportion of upregulated (red) and downregulated (blue) genes (absolute fold change > 1.25). Enrichment P-values were derived from a hypergeometric test, after adjustment for multiple testing (Benjamini and Hochberg’s method).”

9. The authors concluded that ChAdOx1’s attenuation of pro-inflammatory pathways’ activity led to lower disease severity. Yet in the cohort, NAAT+ve ChAdOx1 group has more “moderate A” cases compared to the NAAT+ve placebo group. Can the authors really conclude that they have found a “mechanistic understanding” of how ChAdOx1 reduces disease severity (lines 683-684), when their cohort doesn’t appear to show reduced disease severity in ChAdOx1 group?

The difference between mild and moderate (A) disease is based on subjective symptom reporting (supplementary table 1: lethargy, mild chest tightness and mild SOB on exertion is present in “moderate A” disease but absent in mild disease). Objective measures i.e. RR, HR and SpO2 are the same in mild and moderate A. The subjective nature of symptom reporting introduces noise into the data which means it is wrong to conclude that people in the NAAT+ve ChAdOx1 nCoV-19 group were sicker than those in the NAAT +ve placebo group, particularly given the number of moderate and mild cases in each group were not statistically significantly different. In stage I Chi-squared significance result comparing proportion of mild and moderate cases in the NAAT +ve ChAdOx1 nCoV-19 and placebo group was $p=0.08$, in stage II it was $p=0.44$, pooling stage I and stage II $p=0.17$. In addition, real world data shows that ChAdOx1 nCoV-19 reduces disease severity. It is a strength of our study that we detect this amelioration at the molecular level, despite not being able to detect it at the disease categorisation level – this reflects high the sensitivity of gene expression profiles in capturing disease severity. It supports our point in the conclusion that gene expression studies have the power to identify biomarkers of vaccine effectiveness against severe disease in a relatively small number of people which is useful, especially in early phase clinical studies.

10. The study only looked at symptomatic infections, and for good reason. Could the authors discuss whether this selection can lead to collider bias, for example with the help of a directed acyclic graph? Can the findings in the study be generalized to asymptomatic infections?

Our understanding of collider bias is that it occurs when an exposure and outcome influence a third variable which is then controlled for in study design or analysis. Our differential expression analyses only control for pairing and, in the case of sRNA sequencing red cell counts, which we do not feel constitute potential colliders. We have controlled for risk factors via matching, not by limiting our cohorts to people with particular risk factors. In this sense, we have used a pragmatic approach to cohort selection and analysis which should minimise the potential for collider bias.

We cannot comment on whether the findings in the study be generalized to asymptomatic infections because we do not include such people in our analyses. Given the key issue with COVID-19 is its associated morbidity and mortality, and the benefit of ChAdOx1 nCoV-19 is that it ameliorates this, we feel asking a question about asymptomatic infections, although potentially academically interesting is not of clear clinical relevance. In addition, the presence of inflammation in asymptomatic infections is not well understood. Typically, infectious symptoms are usually driven by an inflammatory response (immune suppressed people can have sepsis but show very little infection symptoms for example) so the fact that an infection is asymptomatic suggests that there is little inflammation in asymptomatic COVID-19 infections, and therefore ChAdOx1 nCoV-19 may have little effect.

Minor comments

1. The authors show a box plot in Fig 7f comparing minimum Ct from SARS-CoV-2 RT-PCR in Figure 7 and conclude that no differences exist between groups. What is the p value? Could this conclusion be due to limited sample size? **The p-value (p=0.26) has been added to the legend. As noted in the introduction, larger studies also show a poor relationship between vaccine status and viral load^{12,13}. We have therefore added and referenced this statement to the discussion: “Although our cohort may be underpowered to observe small differences between the vaccine groups, larger studies also show a poor relationship between vaccine status and viral load^{12,13}”**

2. How does the SARS-CoV-2 viral load at CT+7 compare to that at CT within each vaccine arm? **Though swabs were taken viral loads at CT7 were not run this this analysis is not possible.**

In Fig 7f, could the authors separate stage 1 and stage 2 data in their statistical test? **We have added this information into the figure legend.**

3. The authors describe assaying viral load from nose and throat swabs. Given the known variability in viral load between those 2 sites, the authors should provide more details on the

breakdown, and describe how the addressed anatomical sampling site bias between patients/replicates. I think the reviewer may have assumed that different sites were used for different subjects but in fact a single swab was used to swab the nose and throat for everyone and then placed into media for PCR. The issue between the variability in site is therefore intrinsically controlled for amongst subject, removing the concerns of bias. We have updated the methods section to make this clearer.

4. The authors interchange the terms PCR and NAAT and should choose one. RT-PCR is the type of NAAT test that was used on this sub-study of participants – we have updated the text to make this clearer.

5. A different abbreviation for test visit instead of CT would be better, because CT is usually associated with qPCR cycle threshold. We feel the term CT meaning Covid-19 test visit is clearly defined in our manuscript and have labelled cycle threshold as Ct as per next-gen scientific abbreviation.

6. The plot title of Fig 3d (NAAT+ve (control)) doesn't match the panel label (NAAT+ve (placebo)). We have updated the title to say placebo

7. Please describe how GSEA was done in the Methods section. We have added a methods section on this.

8. A different color choice for the 2 NAAT +ve groups in Fig 3a would be better, in order to avoid confusion with the up- and downregulation colors in the other Fig 3 panels.

The blue and red colours for each of the vaccine groups have been introduced and widely used across all the ChAdOx1 nCoV-19 vaccine trial publications. These colours are consistently used throughout all the figures in the current study, not only figure 3, and clearly depicted and defined in the Figure 2 study design. This should avoid the confusing the reader.

9. A lot of the gene annotations overlap, making them difficult to read, eg, Fig 3e, 7a. We have amended the gene annotations of the volcano plots to prevent overlaps.

10. Please describe what "s0" in Supp Tables 5 & 6 mean in the table captions. S0 was a deprecated term for CT – we have replaced S0 with CT the table accordingly

11. Unintended new line between lines 123 & 124. Heading on line 125 (RNA extraction) is a subheading of line 124 (RNA sequencing)

12. Line 344 should be referring to Supp Fig 1e. Line 344 on our copy refers to the MX1-201 isoform and does not relate to supplementary figure 1 e.

13. Fig 5a and 5b should have the same axis limits for easier comparison. We have removed figure 5b as suggested by another reviewer as there were no differentially expressed genes.

14. Fig 5a, b's titles should say NAAT+ve instead of PCRpos for consistency. We have amended as suggested

15. The horizontal dashed lines in cytokine plots should be explained in the figure captions. Legend updated with explanation

16. Please describe the blood cell count data in the Methods section. Method added to method section.

17. The link on line 443 doesn't work. Many thanks for pointing this out. The link works on our copies. It will be retested at the proofed copy stage.

18. Which figure were the authors referring to on lines 421-423? We have not provided a figure for this. **We have added supplementary figure 16 to illustrate the statement the reviewer is referring to.**

19. There are multiple grammatical issues with respect to the terms COVID-19 versus SARS-CoV-2. For instance, just in the abstract, the authors refer to COVID-19 disease, which is grammatically duplicative; they state that immune responses are measured in response to COVID-19 but rather it should be to SARS-CoV-2 or to the vaccine, and COVID-19 infection is incorrect- it should be SARS-CoV-2 infection.

Many thanks for pointing out the duplicative nature of using the term "COVID-19 disease". We have amended all instances to simply COVID-19. We use the term COVID-19 because our study relates to molecular signatures in the disease (COVID-19) caused by SARS-CoV-2 infection. It is more specific than SARS-CoV-2 infection per se (which would include non-symptomatic infection).

20. Typo in Figure 1 legend 'Participants were then vaccination with either (should be vaccinated). **Typo corrected.**

Reviewers' Comments:

Reviewer #1:

Remarks to the Author:

The revisions implemented have addressed multiple aspects yet we are still unsatisfied with the overall quality of the revised paper.

- It would have been beneficial for the authors to track changes in the main text or use a distinct color for altered content, facilitating a more effective review process.

- The main and supplementary figures' content and quality remain subpar, significantly impacting the manuscript's overall quality. Rather than relying on the reviewers to reiterate these deficiencies once again, the authors should address the previous formatting comments by uniformly applying them to all figures. Many of these issues would likely not be caught and fixed by subsequent editorial processes which would greatly impact the quality of the paper.

- Considering the intricate and complex study design, we also stress again the importance of consistent labeling across ALL figures and the necessity of indicating sample sizes for each group analyzed in each experiment/figure in the legend of every figure in alignment with Supplementary Table 3. This is clearly requested in the guidelines for authors.

- The expectation that vaccine arm allocation for the NAAT-ve group is a major claim and no evidence is presented to support it. We don't know if differential expression have been tested for the two groups and this could easily be done. This is a major point that needs to be addressed given that both groups are treated as one throughout the paper.

- Statistical significance indeed is not impacted the same way by sample size but a trend is expected as error estimates of fold change are impacted by sample size. Only subsampling of the larger group and rerunning the test can support the claim to exclude the possibility of lack of power in the stage 1 dataset.

- The signal in Figure 4a is quite noisy. How is the claim (lines 360-361) that "Seven days after participants presented with COVID-19 symptoms (CT+7), global gene expression trajectories returned towards baseline (Figure 4a)" is supported by this analysis/figure?

- In lines 319-322, the authors argue that COVID cases, symptomatic controls and healthy controls cluster separately in Figure 3a. Given the overlapping points in the figure, this is an inflated conclusion and the wording should be toned down.

- The section from line 381 states that "Direct comparison of the ChAdOx1 nCoV-19 and placebo vaccine groups at CT identified 5 DEGs. GSEA of this comparison showed....". How is GSEA enrichment analysis done with only 5 DEGs? The same question applies to other statements in this section.

- What does "again differences" in the statement in lines 444-446 "There were again differences between vaccine groups at COVID-19 onset, with 1,131 genes differentially expressed between ChAdOx1 nCoV-19 and placebo vaccine recipients at CT" refers to?

Reviewer #2:

Remarks to the Author:

The authors have satisfactorily responded to all comments.

The manuscript is now acceptable for publication.

They have done an excellent job to revise the manuscript. It is particularly acknowledged that they

included a large set of tables showing all DEGs and their statistics.

I still think that volcano plots should better show adjusted p values to avoid confusion with DEGs selection. However, the authors have argued well to follow their line of presenting non-adjusted p values, which I can accept.

My comment to line 675: my mistake, please ignore.

A few very minor comments for which I do not see a revision: In the legend of figure S9, please make reference to the supplement pathway table. In some of the figure legends (e.g. Fig. 7), there is a mix of capital and non-capital letters referring to sub-parts.

Reviewer #3:

None

Reviewer #4:

Remarks to the Author:

Our comments have been largely addressed by the authors.

Minor comments:

1. The figures are unevenly formatted and prepared, leaving the reader with an impression of sloppiness and lack of attention to detail that detract from the overall important findings, these should be improved for the final revision.
2. Could the authors clarify what the yellow circular arrows in Figures 5b, 6e, etc., represent?
3. Lines 601-602: we suggest rephrasing as "ChAdOx1 nCoV-19 vaccination attenuated the activation of immune responses associated with COVID-19 severity." The current phrasing could be understood as the markers themselves being downregulated in ChAdOx1 group, rather than the actual observations that there are fewer changes in mRNA and sRNA expression.
4. Please describe in the Methods section how effect sizes were calculated.
5. Please add the Chi-squared test results mentioned in the authors' response to our previous comment 9 to the appropriate Supplementary tables.
6. Supp Fig 19: The text in panels b & c are too small and difficult to read. The right panel in c has duplicated "Sub analysis".
7. Supplementary tables 5 and 6 still contain the term "s0".

Reviewer #5:

Remarks to the Author:

REVIEWER COMMENTS

Reviewer #1 (Remarks to the Author):

The revisions implemented have addressed multiple aspects yet we are still unsatisfied with the overall quality of the revised paper.

- It would have been beneficial for the authors to track changes in the main text or use a distinct color for altered content, facilitating a more effective review process.

We apologise for not offering a copy with tracked changes, the changes were highlighted in colour instead. We made so many adjustments in the previous revision that the tracked changes version was difficult to read. We have included tracked changes on the most recent version.

- The main and supplementary figures' content and quality remain subpar, significantly impacting the manuscript's overall quality. Rather than relying on the reviewers to reiterate these deficiencies once again, the authors should address the previous formatting comments by uniformly applying them to all figures. Many of these issues would likely not be caught and fixed by subsequent editorial processes which would greatly impact the quality of the paper.

The resolution of the figures in the original word document, which we upload, is high, thus, we assume that the quality is reduced during the upload/conversion process. We like to put our figures into word documents to make it easier for reviewers to read through the manuscript without having to jump to png copies of the final figures. We have replaced a few figures which had lower resolution and readability in the previous version. However, we appreciate figure quality can be limited in word documents and thus frustrating for reviewers. We will be attaching the .png files or other suitable format of each figure to be used at the copy-editing stage. We brought this to the attention of the editor.

- Considering the intricate and complex study design, we also stress again the importance of consistent labeling across ALL figures and the necessity of indicating sample sizes for each group analyzed in each experiment/figure in the legend of every figure in alignment with Supplementary Table 3. This is clearly requested in the guidelines for authors.

We have added group sizes to all the figure legends in addition to providing that information in Supplementary table 3.

- The expectation that vaccine arm allocation for the NAAT-ve group is a major claim and no evidence is presented to support it. We don't know if differential expression have been tested for the two groups and this could easily be done. This is a major point that needs to be addressed given that both groups are treated as one throughout the paper.

We understand the reviewer comment to mean "The expectation that vaccine arm allocation is not biologically relevant for the NAAT-ve group is a major claim and no evidence is presented to support it". Apologies if this is not what was meant.

We have now included the output of a differential expression analysis showing that there were no differences between NAAT-ve participants based on vaccine group. This information is contained in Supplementary figure 4 and referred to in the text as follows: "There were no differences in the transcriptome between ChAdOx1 nCoV-19 vaccinees and placebo recipients at CT for the NAAT-ve group (Supplementary figure 4)."

- Statistical significance indeed is not impacted the same way by sample size but a trend is expected as error estimates of fold change are impacted by sample size. Only subsampling of the larger group and rerunning the test can support the claim to exclude the possibility of lack of power is the stage 1 dataset.

Statistical significance is impacted by sample size, so we do not fully understand the word “not” in reviewers’ statement “Statistical significance indeed is not impacted the same way by sample size”.

We think the reviewer is asking us about power in the Stage 1 cohort. Apologies if we have misunderstood. We ran a power calculation to estimate the power in the Stage 1 cohort using ssizeRNA package and attached the resulting plots as a Supplementary figure 22 k and l.

- The signal in Figure 4a is quite noisy. How is the claim (lines 360-361) that “Seven days after participants presented with COVID-19 symptoms (CT+7), global gene expression trajectories returned towards baseline (Figure 4a)” is supported by this analysis/figure?

The text has been amended accordingly to explain how the result is supported by the figure: *Seven days after participants presented with COVID-19 symptoms (CT+7), global gene expression trajectories returned towards baseline with CT+7 samples predominantly moving to the bottom of the plot compared to CT samples and located in the proximity to D0 (Error! Reference source not found.a).*

Following explanation of the arrows has been added to the Figure 4a legend: *The dashed arrows connect the different time points from the same participants; the direction of the arrows is pointing towards the latter time point.*

- In lines 319-322, the authors argue that COVID cases, symptomatic controls and healthy controls cluster separately in Figure 3a. Given the overlapping points in the figure, this is an inflated conclusion and the wording should be toned down.

We agree that samples do not cluster 100% discretely (i.e. there are some overlap), therefore we removed that sentence and rewrote the first part of the paragraph as follows:

In the stage 1, we investigated the distribution and relationships between the samples from different study groups (Error! Reference source not found.a). Gene set enrichment analysis revealed significantly enriched pathways between COVID-19 cases, symptomatic controls (i.e. COVID-19 negative cases), and health (baseline) (Error! Reference source not found.a-f), reflecting distinct next-gen RNA blood transcriptome profiles.

- The section from line 381 states that “Direct comparison of the ChAdOx1 nCoV-19 and placebo vaccine groups at CT identified 5 DEGs. GSEA of this comparison showed.....”. How is GSEA enrichment analysis done with only 5 DEGs? The same question applies to other statements in this section.

We refer the reader to our methods section on GSEA. We think the reviewer believes us to have done a gene set analysis using an over representation analysis approach (ORA) which looks at what pathways are overrepresented amongst differentially expressed genes. We used a GSEA approach best known as the broad approach. It looks at which pathways are enriched amongst genes towards the top and bottom of whole list of analysed genes (regardless of whether their p-value was <0.05). Indeed, you do not need to have any differentially expressed genes to run this analysis. GSEA methodology is described in the Methods section. For more information, please see <https://doi.org/10.1073/pnas.0506580102> paper.

We also copy our explanation to one of the previous reviewers here in case this was not seen by reviewer 1. We are assuming that the reviewer is asking how a pathway can be enriched yet genes within that pathway not reach the significance threshold for differential expression. The GSEA gene module analysis uses the entire gene list output of a differential expression analysis

with that gene list ranked by the t-test statistic (regardless of whether a gene's p-value was less than 0.05). This type of analysis is very powerful because it is based on the overall pattern of where a pathway's genes fall within that list. For example, if the majority of a pathway's genes lie at the top of the list – i.e. are upregulated (even if some/all of those genes are not called differentially expressed) then that pathway is positively enriched. There can be some discrepancies between the enrichment results for the stage I and II data – this may be partially explained by the larger sample size in the stage II data which means that the ranking of a gene within a list is less subject to random fluctuation arising from the impact of a single/few random extreme sample values which can have a large influence when dealing with small sample sizes. Sample size limitations discussed in discussion.

- What does “again differences” in the statement in lines 444-446 “There were again differences between vaccine groups at COVID-19 onset, with 1,131 genes differentially expressed between ChAdOx1 nCoV-19 and placebo vaccine recipients at CT” refers to?
We have swapped the word "again" to "as in the stage 1 cohort".

Reviewer #2 (Remarks to the Author):

The authors have satisfactorily responded to all comments.

The manuscript is now acceptable for publication.

They have done an excellent job to revise the manuscript. It is particularly acknowledged that they included a large set of tables showing all DEGs and their statistics.

I still think that volcano plots should better show adjusted p values to avoid confusion with DEGs selection. However, the authors have argued well to follow their line of presenting non-adjusted p values, which I can accept.

My comment to line 675: my mistake, please ignore.

A few very minor comments for which I do not see a revision: In the legend of figure S9, please make reference to the supplement pathway table. In some of the figure legends (e.g. Fig. 7), there is a mix of capital and non-capital letters referring to sub-parts.

We thank the reviewer for their valuable input and positive comments. Minor comments have been addressed with figure legends corrected and a supplementary results file has been added as Supplementary table 14 that contains the names of the pathways in Figure S9. We have updated the text to refer the reader to this.

Reviewer #4 (Remarks to the Author):

Our comments have been largely addressed by the authors.

Minor comments:

1. The figures are unevenly formatted and prepared, leaving the reader with an impression of sloppiness and lack of attention to detail that detract from the overall important findings, these should be improved for the final revision.

The resolution of the figures in the original word document, which we upload, is high, thus, we assume that the quality is reduced during the upload/conversion process. We like to put our figures into word documents to make it easier for reviewers to read through the manuscript without having to jump to png copies of the final figures. We have replaced a few figures which had lower resolution and readability in the previous version. However, we appreciate figure quality can be limited in word documents and thus frustrating for reviewers. We will be attaching the .png files or other suitable format of each figure to be

used at the copy-editing stage. We brought this to the attention of the editor. **Irina – note/remind Daniel when he submits**

2. Could the authors clarify what the yellow circular arrows in Figures 5b, 6e, etc., represent?

The following explanation has been added to the legends of all the relevant figures and supplementary figures: *Yellow circular lines represent the direction of changes observed compared to baseline – left to right in the upper part of the plot represents upregulation and right to left in the bottom part of the plot represents downregulation.*

3. Lines 601-602: we suggest rephrasing as “ChAdOx1 nCoV-19 vaccination attenuated the activation of immune responses associated with COVID-19 severity.” The current phrasing could be understood as the markers themselves being downregulated in ChAdOx1 group, rather than the actual observations that there are fewer changes in mRNA and sRNA expression.

This sentence has been replaced as suggested.

4. Please describe in the Methods section how effect sizes were calculated.

The description of the effect sizes calculations has been now added to the Methods section

5. Please add the Chi-squared test results mentioned in the authors’ response to our previous comment 9 to the appropriate Supplementary tables.

This has been now added to the footnotes of the Supplementary tables 5 and 6.

6. Supp Fig 19: The text in panels b & c are too small and difficult to read. The right panel in c has duplicated “Sub analysis”.

The figure has been updated accordingly and replaced.

7. Supplementary tables 5 and 6 still contain the term “s0”.

Thank you for noticing, this has been corrected in both tables.

Reviewer #5 (Remarks to the Author):
